# Dissecting aggregation and seeding dynamics of α-Syn polymorphs using the phasor approach to FLIM

Jessica Tittelmeier [1,2], Silke Druffel-Augustin[1], Ania Alik[3], Ronald Melki [3] & Carmen Nussbaum-Krammer [1,2 ✉]

Synucleinopathies are a heterogenous group of neurodegenerative diseases characterized by the progressive accumulation of pathological α-synuclein (α-Syn). The importance of structural polymorphism of α-Syn assemblies for distinct synucleinopathies and their progression is increasingly recognized. However, the underlying mechanisms are poorly understood. Here we use fluorescence lifetime imaging microscopy (FLIM) to investigate seeded aggregation of α-Syn in a biosensor cell line. We show that conformationally distinct α-Syn polymorphs exhibit characteristic fluorescence lifetimes. FLIM further revealed that α-Syn polymorphs were differentially processed by cellular clearance pathways, yielding fibrillar species with increased seeding capacity. Thus, FLIM is not only a powerful tool to distinguish different amyloid structures, but also to monitor the dynamic process of amyloid remodeling by the cellular environment. Our data suggest that the accumulation of highly seeding competent degradation products for particular polymorphs may account for accelerated disease progression in some patients.

[1] Center for Molecular Biology of Heidelberg University (ZMBH) and German Cancer Research Center (DKFZ), DKFZ-ZMBH Alliance, Heidelberg, Germany.
[2] Department of Anatomy II - Neuroanatomy, Ludwig-Maximilians-University of Munich, Munich, Germany. [3] Institute Francois Jacob (MIRCen), CEA, and Laboratory of Neurodegenerative Diseases, CNRS, Fontenay-Aux-Roses, France. ✉email: carmen.nussbaum@med.uni-muenchen.de

The accumulation and prion-like propagation of amyloid deposits is a hallmark of many neurodegenerative diseases. Aggregated SNCA/α-synuclein (α-Syn) is associated with diseases termed synucleinopathies, including Parkinson's disease and multiple system atrophy. Despite their common cause, synucleinopathies are highly heterogeneous. These different pathologies may be due to the fact that α-Syn forms conformationally distinct amyloid assemblies[1–5]. Indeed, distinct α-Syn polymorphs have been shown to differ in their aggregation and propagation propensities and levels of toxicity in vivo[1–3]. Moreover, the cellular environment seems to play a crucial role[6].

To decipher the progressive accumulation and spreading of disease-related proteins, the underlying mechanisms need to be studied in more detail in the cellular context. Biochemical assays have been widely used to analyze aggregation, but they are often too indirect and require the extraction of aggregated proteins from whole cell or animal lysates, which could lead to artifacts. Imaging techniques can be used to study aggregation in situ without the need for cellular lysis.

Methods such as fluorescence recovery after photobleaching and fluorescence loss in photobleaching have been used in an attempt to improve aggregate characterization. Unfortunately, these techniques only distinguish between mobile and immobile aggregation states, but clearly, there are more aspects to an aggregate, such as the specific packing of the individual monomers in an amyloid fiber that remain invisible.

Aggregation into an amyloid fiber leads to quenching of an attached fluorophore due to compaction and crowding, thus reducing its fluorescence lifetime[7,8]. This can be measured using fluorescence lifetime imaging microscopy (FLIM)[7–11]. Therefore, FLIM is increasingly used to investigate aggregation processes in various models[12–16]. However, interpretation of fluorescence lifetimes becomes challenging when resolving lifetimes requires tedious multiexponential decay curve fitting, which is often the case with biological samples[17]. Here we use the phasor plot method to visualize the complex nature of aggregation in a fit-free manner. Each pixel of an intensity image is mapped to a point in the phasor plot corresponding to the mean of the measured fluorescence lifetime[18]. Single exponential lifetimes and complex decays lie on the universal circle or within, respectively, with longer lifetimes located the closest to the (0,0) coordinate, while shorter lifetimes are shifted toward the right[19]. This approach provides a convenient and fast global view of the fluorescence lifetime distribution, and unveils the high complexity of amyloid aggregation dynamics in the cellular environment.

In this study, we utilized the phasor approach to FLIM to monitor seeded aggregation of an α-Syn reporter construct in cell culture following the introduction of in vitro generated α-Syn fibrils. We show that different α-Syn polymorphs display distinct signatures in FLIM, which reflects their distinct folds as well as individual interactions with the cellular environment. FLIM also revealed that the seeds are getting processed by the cellular protein quality control network over time, as evidenced by a gradual increase in fluorescence lifetime. However, instead of rapid solubilization or degradation, α-Syn species with intermediate fluorescence lifetimes accumulated that preferentially localized to the seeded endogenous α-Syn aggregate. Although some polymorphs were differentially susceptible to certain protein clearance pathways, blocking their cellular processing reduced the formation of these intermediates as well as endogenous α-Syn aggregates. Thus, incomplete cellular clearance of the amyloid seed accelerates seeded aggregation of native α-Syn, which could play a role in the progressive spreading of disease pathology with age. We further conclude that FLIM provides an additional window to features that remain hidden with conventional intensity-based imaging techniques.

## Results

**Monitoring seeded aggregation of α-Syn by phasor-FLIM.** To evaluate the use of the phasor approach to FLIM to investigate seeded aggregation of α-Syn, a biosensor HEK cell line expressing the disease-related aggregation prone A53T mutant α-Syn, fused to yellow fluorescent protein (α-SynA53T-YFP)[20] was treated with different in vitro assembled α-Syn fibers (Supplementary Fig. 1a). Under normal growth conditions, these cells display mainly a diffuse α-SynA53T-YFP signal (Fig. 1a). Accordingly, the fluorescence lifetime of α-SynA53T-YFP is very homogenous, as seen in the phasor where the relative occurrence of fluorescence lifetimes is plotted. Addition of Fibrils resulted in the formation of distinct cytosolic foci in the fluorescence intensity image and a shift towards shorter α-SynA53T-YFP lifetimes in the corresponding phasor plot (Fig. 1a). In the phasor plots, a specific region of interest (ROI) can be selected to highlight pixels with that particular lifetime in the corresponding fluorescence intensity image[21]. Selecting an ROI around shorter lifetimes (purple cursor) in the phasor plot reveals that these pixels colocalize with foci (Fig. 1b). Due to the reciprocity between intensity image and phasor plot, selecting an ROI around foci in the intensity image leads to the selective visualization of these pixels in the phasor plot, confirming that the lifetime of fluorophores within foci is shortened (Supplementary Fig. 1b). Accordingly, in Fibril-seeded cells a shift in the fluorescence lifetime distribution towards shorter lifetimes (Fig. 1c) and a significant decline in the total mean fluorescence lifetime of α-SynA53T-YFP was detected (Fig. 1d).

α-Syn can form amyloid assemblies with distinct structures, so-called conformational polymorphs, which exhibit distinct pathological characteristics in vitro and in vivo[22,23]. During prion-like propagation the distinct conformations are often imposed on native α-Syn by templated seeding[24,25]. Therefore, it was of interest to investigate whether the conformation of the fibrillar α-Syn seeds could have a measurable effect on the fluorescence lifetime of the seeded α-SynA53T-YFP aggregates. In addition to Fibrils, we seeded the biosensor cell line with other structurally well-characterized fibrillar α-Syn polymorphs, F65, F91, and Ribbons[22,23,26,27]. These polymorphs differ in the amino acids located in their core and those exposed at their surface, resulting in a distinct fiber architecture (Supplementary Fig. 1a)[22,23,26,27]. Similar to Fibrils, F65 and F91 polymorphs induced the formation of α-SynA53T-YFP foci. This led to a decreased mean fluorescence lifetime of α-SynA53T-YFP in seeded compared to non-seeded cells (Fig. 1e–g, Supplementary Fig. 1c). Ribbons had the lowest seeding capacities of the different polymorphs in our experimental model (Supplementary Fig. 1c), which was reflected by an unchanged fluorescence lifetime distribution and no significant difference in the mean lifetime of α-SynA53T-YFP compared to the control (Fig. 1e–g). Selection of short-lived α-SynA53T-YFP species on the phasor plot in Ribbon-seeded cells revealed that they localized to foci (Fig. 1e). However, these foci also contained α-SynA53T-YFP species with longer fluorescence lifetimes. Therefore, α-SynA53T-YFP foci seeded by Ribbons have a significantly higher mean fluorescence lifetime than α-SynA53T-YFP foci seeded by Fibrils, F65, or F91 polymorphs (Fig. 1e, Supplementary Fig. 1d). No robust difference in the mean fluorescence lifetimes of α-SynA53T-YFP in foci seeded with Fibrils, F65 or F91 polymorphs could be detected (Supplementary Fig. 1d). Hence, the seeded aggregation of endogenous α-SynA53T-YFP by the addition of exogenous α-Syn polymorphs Fibrils, F65, and F91 led to the accumulation of short-lived protein species. However, the conformation of the added seeds did not cause a significant difference in the respective fluorescence lifetimes.

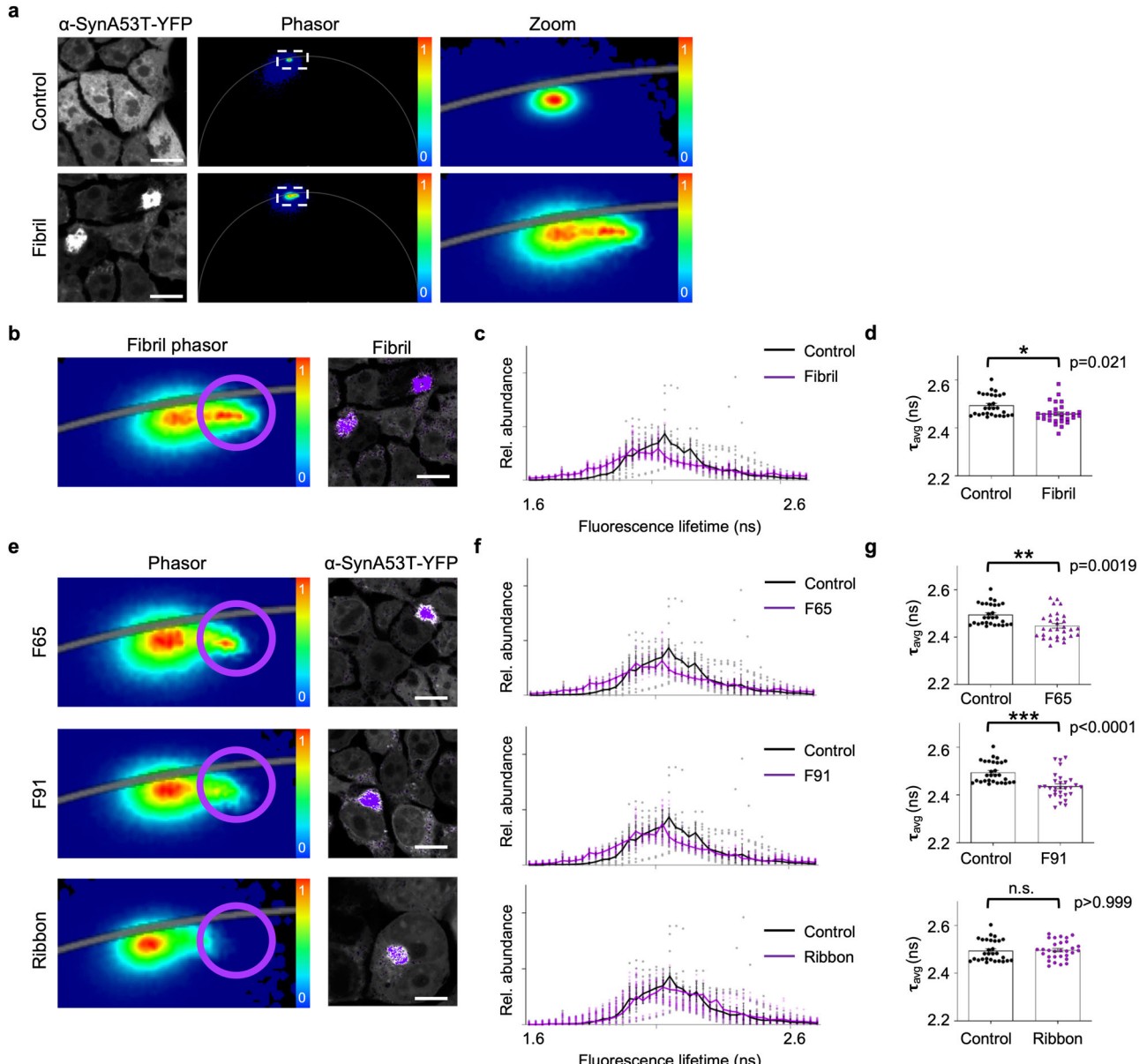

**Fig. 1 FLIM analysis of seeded aggregation of α-SynA53T-YFP using distinct α-Syn polymorphs. a** Fluorescence intensity images of HEK cells expressing α-SynA53T-YFP seeded with indicated α-Syn fibers. The phasor displays the relative occurrence of lifetimes in the accompanying image (0–1). Although most of the signal is from soluble α-SynA53T-YFP, the corresponding phasor plots of YFP fluorescence lifetimes show a shift in the distribution towards shorter lifetimes with seeded aggregation. **b, e** Choosing a region of interest (ROI, purple) around shorter lifetimes in the phasor plot generates a pseudo colored image in which each pixel with a lifetime within the ROI is colored according to the color of the cursor in the phasor plot (purple cursor). The position of the cursor is arbitrary. The lifetime distribution of the non-seeded α-SynA53T-YFP in control cells was used as a reference to select the shorter α-SynA53T-YFP lifetimes that occur only in the seeded cells. Shorter lifetimes localize to distinct cytosolic foci. **c, f** Fluorescence lifetime histograms of α-SynA53T-YFP seeded with indicated α-Syn polymorphs, averaged over all acquired images ($n = 30$ for each polymorph, mean ± SEM). Seeding with α-Syn polymorphs resulted in a small shoulder of shorter lifetimes in the overall distribution compared to the lifetime distribution of α-SynA53T-YFP in non-seeded cells. **d, g** The mean fluorescence lifetime ($\tau$) of α-SynA53T-YFP in non-seeded control cells compared to cells seeded with indicated polymorphs. Data are shown as mean ± SEM. Statistical analysis was done using Kruskal-Wallis one-way analysis with Dunn's multiple comparison test. Scale bar, 10 μm. Distinct α-Syn polymorphs are able to induce a shift in the α-SynA53T-YFP lifetime distribution towards shorter lifetimes that colocalize with foci.

**The different α-Syn polymorphs show a distinct distribution of fluorescence lifetime in phasor-FLIM.** We wondered whether FLIM was sensitive enough to detect conformational differences between amyloid assemblies and therefore sought to elucidate whether the seeds themselves had different fluorescence lifetimes. To test this, the distinct α-Syn polymorphs were labeled following their assembly with Atto647 for direct visualization (Supplementary Fig. 2a). This not only allows imaging of the seed within a cell

but also monitoring of the fluorescence lifetime of both the endogenous α-SynA53T-YFP and the fibrillar α-Syn seeds (α-Syn-Atto647). We observed that seeding with α-Syn-Atto647 did not cause major differences in the lifetime distribution of α-SynA53T-YFP compared to seeding with unlabeled seeds (Fig. 1, Supplementary Fig. 2b–d). In contrast, there were massive differences between the lifetimes of various α-Syn-Atto647 polymorphs (Fig. 2). Pseudo coloring of the lifetime distribution on the phasor

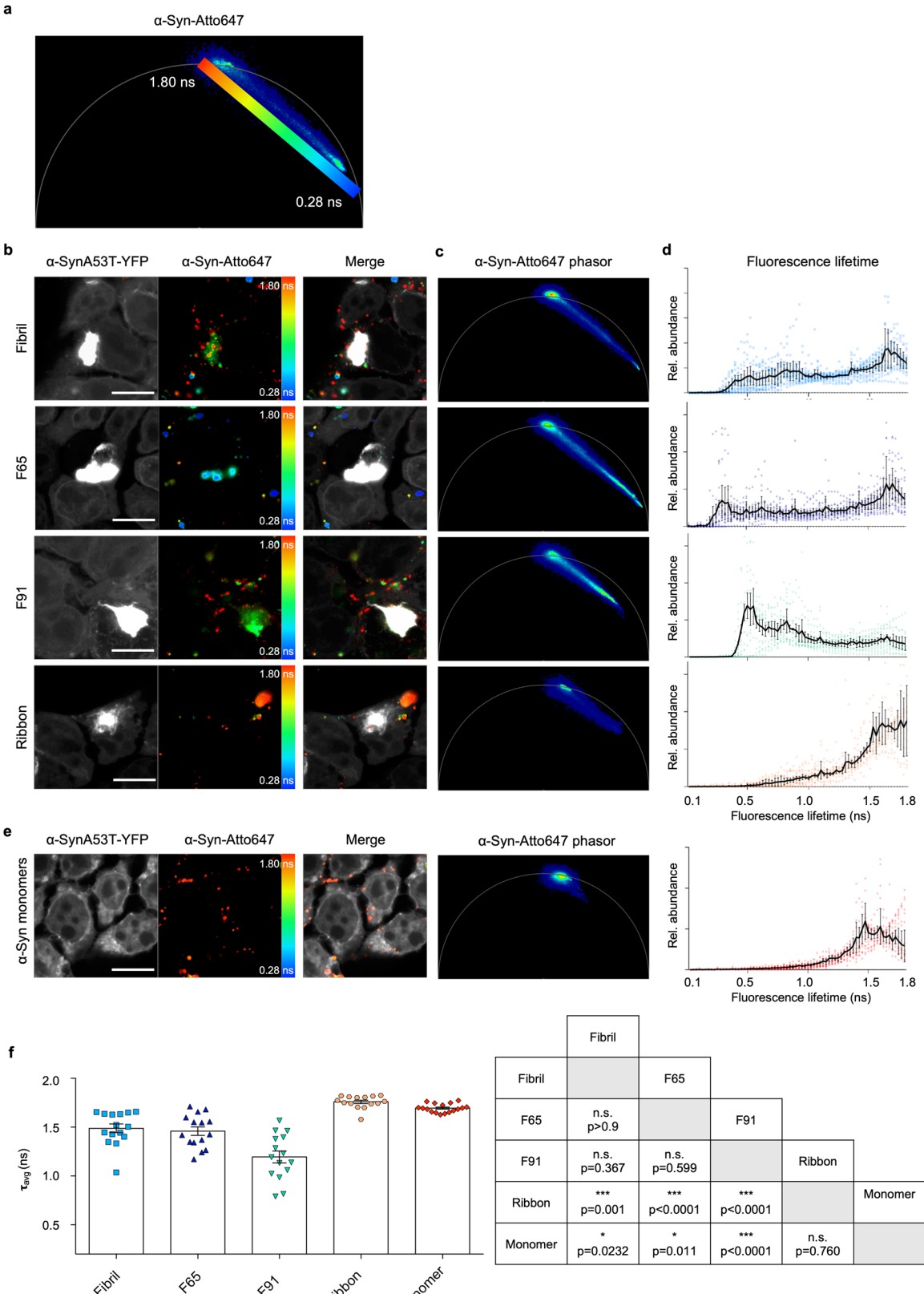

map was used to highlight the location of shorter-lived species (blue) and green, yellow, and red species representing their progressively longer-lived fluorescence lifetimes in the corresponding fluorescence intensity image (Fig. 2a, b). Notably, Fibril, F65, and F91 polymorphs displayed a broad range of fluorescence lifetimes, ranging from around 0.2 ns to 1.8 ns (Fig. 2c, d). F65 seeds formed species with the shortest fluorescence lifetime that do not appear with Fibrils and F91, while F91 formed species with a fluorescence lifetime of no less than 0.5 ns. Conversely, Ribbons did not exhibit such a wide distribution of fluorescence lifetimes, with the majority of seeds exhibiting a mean lifetime of 1.7 ns, which was similar to the mean lifetime of monomeric α-Syn in the cellular environment

**Fig. 2 α-Syn polymorphs exhibit signature phasor fingerprints. a** Pseudo coloring LUT from short fluorescence lifetime (0.28 ns) in blue to longer fluorescence lifetime (1.8 ns) in red on the phasor plot used in all α-Syn-Atto647 pseudo colored images. **b** Fluorescence intensity image of α-SynA53T-YFP and α-Syn-Atto647 seeds with pseudo colored from blue (0.28 ns) to red (1.8 ns) with the corresponding merged image. Scale bar, 10 μm. **c** Corresponding phasor plot of the fluorescence lifetimes of α-Syn-Atto647 seeds. **d** Histograms of α-Syn-Atto647 showing the relative abundance (rel. abundance) of pixels exhibiting a certain fluorescence lifetime (ranging from 0 to 1.8 ns), averaged over all acquired images ($n = 15$–$20$ for each polymorph, mean ± SEM). Each α-Syn polymorph shows a signature fluorescence lifetime distribution. **e** Monomeric α-Syn displays a long fluorescence lifetime around 1.8 ns. Histograms of α-Syn-Atto647 monomers showing the relative abundance (rel. abundance) of pixels exhibiting a certain fluorescence lifetime (ranging from 0 to 1.8 ns), averaged over all acquired images ($n = 17$, mean ± SEM). Scale bar, 10 μm. **f** The mean fluorescence lifetime ($\tau$) of α-Syn polymorphs. Data are shown as mean ± SEM. Statistical analysis was done using Kruskal-Wallis one-way analysis with Dunn's multiple comparison test.

(Fig. 2e, f). This indicates that Ribbons are essentially invisible in FLIM and are incorrectly detected as monomeric α-Syn, although they are clearly fibrillar (Supplementary Fig. 2a), which led us to remove them from all subsequent analyses. Nevertheless, all visible α-Syn polymorphs exhibit a characteristic distribution of fluorescence lifetimes when exposed to cells, which is revealed by phasor-FLIM.

**The cellular environment affects the lifetime distribution of α-Syn polymorphs.** We wanted to further investigate the origin of the broad lifetime distributions of Fibril, F65, and F91 and asked whether this distribution is an intrinsic property of the particular polymorph. To test this, the fluorescence lifetime of α-Syn polymorphs was assessed directly. The initial seeds of Fibril, F65, and F91 each exhibited a specific short fluorescence lifetime with only a very narrow distribution (Fig. 3a–c). Therefore, we hypothesized that the broad distribution of fluorescence lifetime (Fig. 2b–d) could be attributed to the intracellular environment. We asked whether it is caused by the interaction of the fluorophore Atto647 with YFP during the seeding process. To test this, fibrillar seeds were added to HEK cells not expressing α-SynA53T-YFP. The α-Syn polymorphs still exhibited their distinct broad distribution of fluorescence lifetimes when introduced into HEK cells (Fig. 3d, Supplementary Fig. 3). Hence, the signature fluorescence lifetime fingerprint of the polymorphs in cells is not dependent on their interaction with α-SynA53T-YFP.

Since the broad distribution of fluorescence lifetimes seems to come from exposure to the cellular environment, we set out to investigate the kinetics behind α-Syn seeds developing longer fluorescence lifetimes. While after 4 hours the internalized Fibril-Atto647 seeds resemble the initial Fibril-Atto647 seeds not exposed to cells, there is a time-dependent accumulation of species with increasingly longer fluorescence lifetimes (Fig. 4a–c, Fig. 3a–c). At 12 h post-seeding, the characteristic broad fluorescence lifetime distribution in the phasor can be observed for the first time. In accordance, the relative abundance of longer fluorescence lifetimes increases in the corresponding lifetime histograms and the mean fluorescence lifetime (Fig. 4b, c). Between 12 h and 24 h after seeding, an increasing number of species with fluorescence lifetimes between 1.5 ns and 1.8 ns form, possibly monomers, given the fluorescence lifetime of monomeric α-Syn. This processing of the seeds over time is also seen for F65 and F91 (Supplementary Fig. 4a, b). We conclude that the large distribution of lifetimes observed for some α-Syn seeds is a result of cellular processing over time. Intriguingly, intermediate fluorescence lifetime species preferentially localize to α-SynA53T-YFP foci, rather than the likely unprocessed original seed with a very short fluorescence lifetime (Fig. 4d). We, therefore, speculated that there might be a correlation between the cellular processing of seeds and the seeding efficiency of α-Syn polymorphs.

**Processing of α-Syn fibrillar polymorphs by protein quality control systems aggravates seeding.** We investigated whether cellular processing of a seed may give rise to a more

seeding-competent species. Several cellular pathways might mediate this processing. In our approach, seeds are delivered directly into the cytosol of the cell by transfection. Three main protein quality control systems are responsible for the cytosolic removal of protein aggregates. These are the autophagy-lysosomal system and the ubiquitin-proteasome system, both of which accomplish protein degradation, or chaperone-mediated disaggregation, which intends to dissolve aggregated proteins. We, therefore, tested the impact of these pathways on the processing of Fibrils. First, we targeted the lysosomal degradation pathway. Chloroquine (CQ) was added 12 h before seeding with Fibrils, to increase lysosomal pH and thus block autophagic flux[28]. 12 h after seeding, the characteristic fingerprint of fluorescence lifetimes developed in the control, but to a lesser extent with lysosomal degradation blocked (Fig. 5a, b). A short-lived fluorescence lifetime was more prominently detected, reminiscent of the initially introduced seed (Fig. 5a, b, Fig. 4a). In accordance, the mean fluorescence lifetime of Fibril-Atto647 is shorter upon CQ treatment (Fig. 5b). Remarkably, the formation of endogenous α-SynA53T-YFP foci also significantly decreased with inhibition of the autophagy-lysosomal system (Fig. 5g, Supplementary Fig. 5a). Thus, the broadening of the fluorescence lifetime distribution in FLIM appears to indeed reflect processing of Fibrils by lysosomal degradation. The results further suggest that this processing generates more seeding-competent species.

To assess the role of the ubiquitin-proteasome system in the processing of Fibrils, we blocked the proteasome for 4 h using MG132 before the addition of Fibrils. Similar to the lysosomal degradation inhibition, this partially impeded the development of the characteristic longer fluorescence lifetimes of Fibrils after 12 h of seeding (Fig. 4c, d). Again, this was accompanied by a reduction in α-SynA53T-YFP foci formation in the cells treated with MG132 (Fig. 5h, Supplementary Fig. 5b). Hence, inhibiting the processing of Fibrils, either by blocking the autophagy-lysosomal system or the ubiquitin-proteasome system, reduced seeded aggregation of α-Syn.

The disassembly of amyloid aggregates by the HSP110/HSP70 disaggregation machinery has been shown to generate smaller seeding competent species[29,30]. Therefore, we wondered whether targeting this disaggregation activity would also prevent the processing of Fibrils and impact α-Syn seeded aggregation. Disaggregation by the HSP110/HSP70 system depends on the substrate recognition and recruitment of HSP70 by the J-domain protein DNAJB1[31,32]. Consequently, we targeted disaggregation by knocking down DNAJB1 prior to seeding (Supplementary Fig. 6a). DNAJB1 knockdown (KD) partially hampered the processing of the original Fibril seeds into species with longer fluorescence lifetimes (Fig. 5e, f). Disaggregation of Fibrils also seemed to generate seeding competent species as there is a reduction in α-SynA53T-YFP foci formation in cells with reduced DNAJB1 levels (Fig. 5i). Of note, while DNAJB1 KD reduced foci formation in general, we noticed an increase in elongated foci as opposed to the typical spherical foci (Supplementary Fig. 5c zoom, 5d), suggesting that DNAJB1 may affect not only exogenously added seeds but also endogenous α-SynA53T-YFP

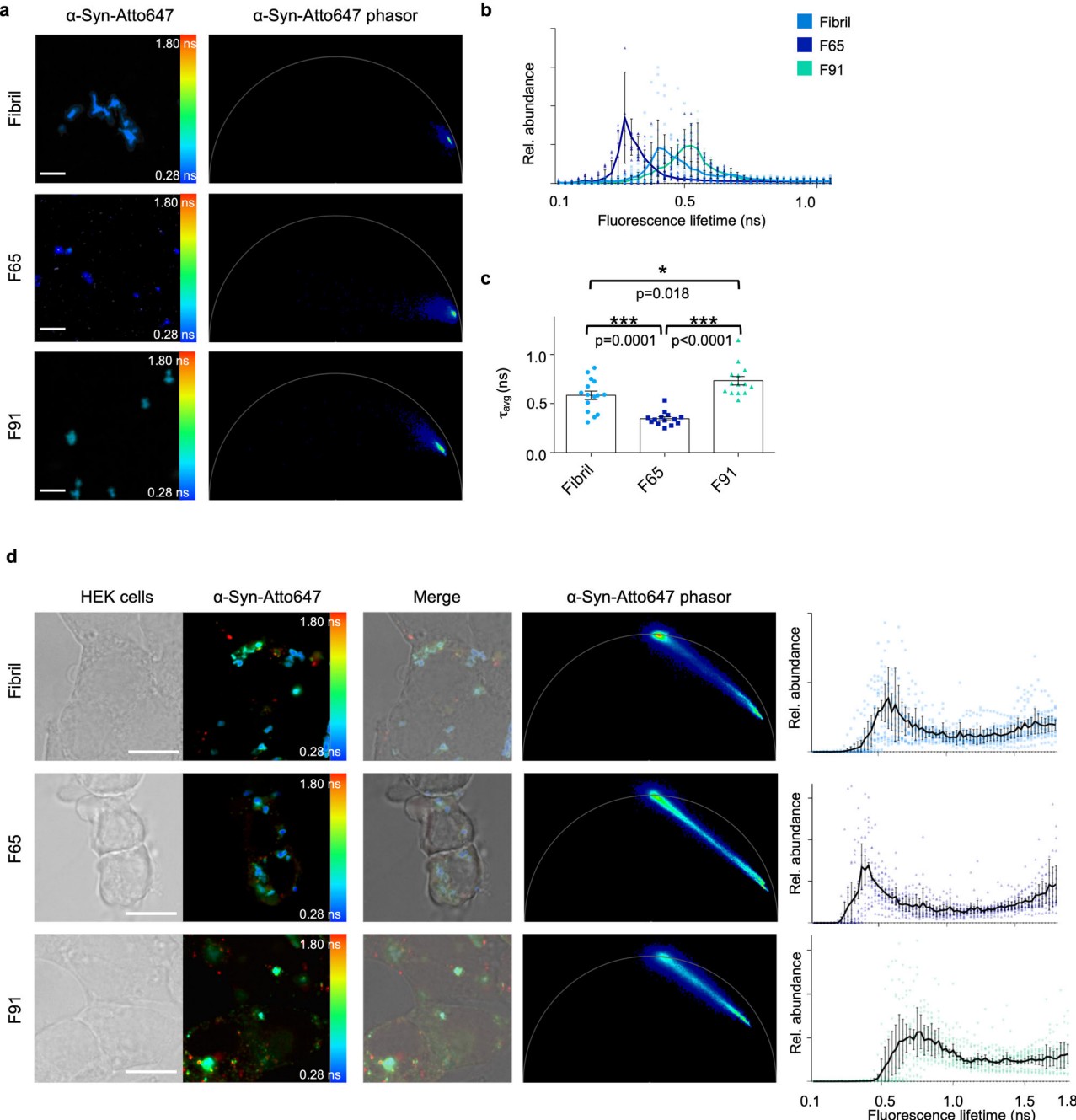

**Fig. 3 The cellular environment affects the lifetime distribution of α-Syn polymorphs. a** Fluorescence intensity image of α-Syn-Atto647 seeds pseudo colored from blue (0.28 ns) to red (1.8 ns) to illustrate short (blue) and long (red) fluorescence lifetimes and the corresponding phasor plots when not added to cells. **b** Histograms of Fibril, F65, and F91 seeds showing the relative abundance (rel. abundance) of pixels with a certain fluorescence lifetime, averaged over all acquired images ($n = 15$ for each polymorph, mean ± SEM). **c** The mean fluorescence lifetime ($\tau$) of α-Syn polymorphs. Data are shown as mean ± SEM. Statistical analysis was done using one-way ANOVA with Turkey's multiple comparison test. Each α-Syn polymorph shows a signature fluorescence lifetime distribution and mean. Scale bar, 5 μm. **d** Transmitted light image of HEK293 cells and fluorescence intensity image of pseudo colored α-Syn-Atto647 seeds when added to cells with the merged image and the corresponding phasor plot of the fluorescence lifetime. α-Syn polymorphs still exhibit a characteristic broadening of lifetime distribution when added to HEK cells not expressing α-SynA53T-YFP. Histograms of α-Syn-Atto647 seeds showing the relative abundance (rel. abundance) of pixels with a certain fluorescence lifetime (ranging from 0 to 1.8 ns), averaged over all acquired images ($n = 15$, mean ± SEM). Scale bar, 10 μm.

aggregates. As a control, we targeted DNAJA2, which is involved in the prevention of aggregation rather than disaggregation[32,33]. In accordance, DNAJA2 KD did not impact the fluorescence lifetime of Fibrils and actually slightly increased their seeding efficiency, in line with a role in preventing α-Syn aggregation (Supplementary Figs. 5e–g, 6b).

When the treatments were repeated with F65 and F91 polymorphs, we found that the different protein clearance pathways could not process all α-Syn polymorphs to a similar extent as Fibrils. Whereas inhibition of the autophagy-lysosomal system blocked the processing of both F65 and F91 polymorphs (Fig. 6a, b, Supplementary Figs. 7a, 8a), which was accompanied

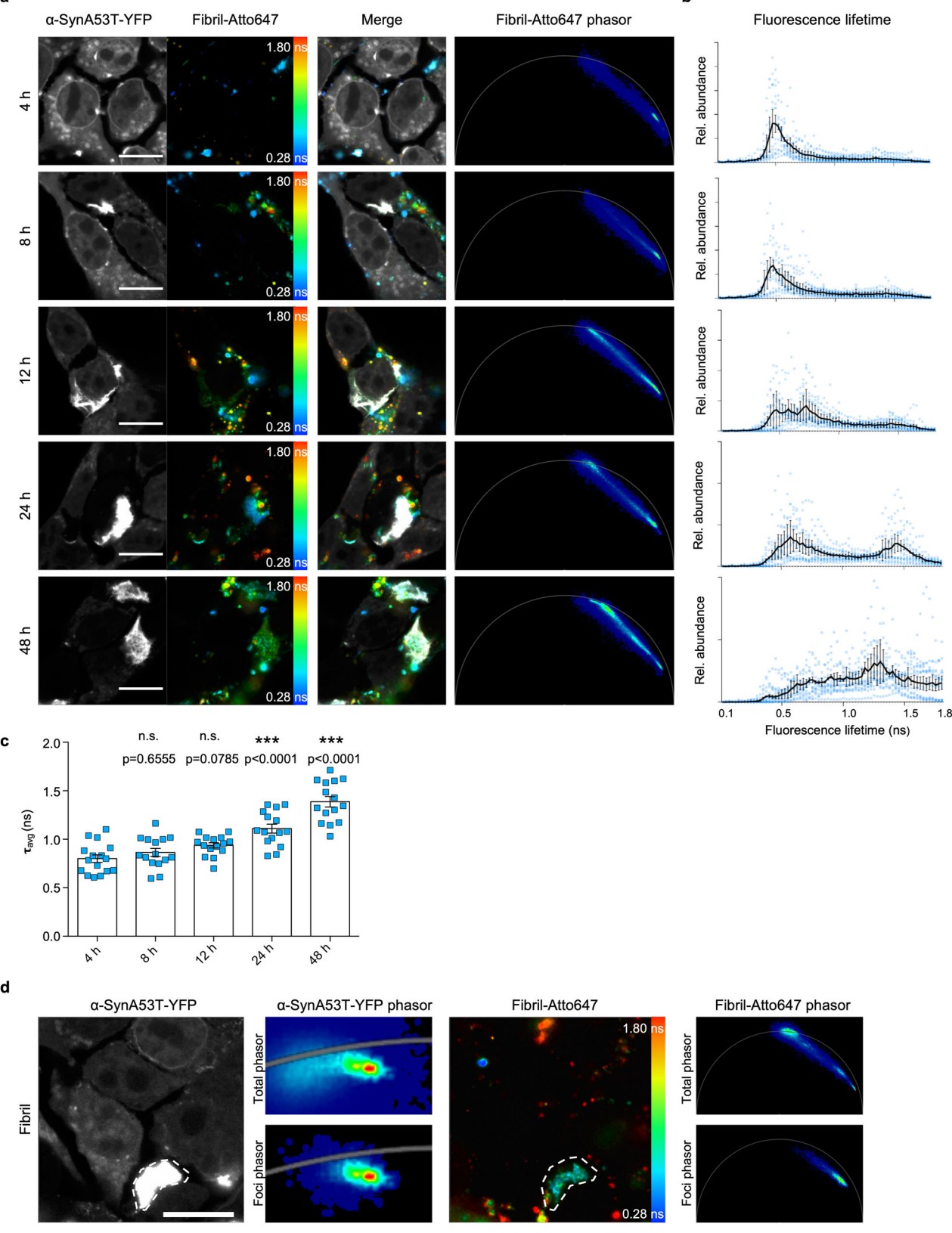

by reduced seeding efficiency (Fig. 6c, Supplementary Figs. 7b, 8b), it differed for the ubiquitin-proteasome pathway. For F65, inhibition of proteasomal degradation diminished the generation of species with longer fluorescence lifetimes, while processing of F91 was only marginally affected by the MG132 treatment. In accordance, there was a reduction in formation of endogenous α-SynA53T-YFP foci only in the case of F65, but not in the case of F91 (Fig. 6d–f, Supplementary Fig. 6c, d, 8c, d). This suggests that the proteasome may not be able to process the initial F91 seeds, as is the case for F65 and Fibril.

Moreover, DNAJB1 KD partially suppressed the processing of the F65 seeds into species with longer fluorescence lifetimes

**Fig. 4 Kinetics of Fibril seed processing. a** Fluorescence intensity images of α-SynA53T-YFP and α-Syn-Atto647 seeds pseudo colored from blue (0.28 ns) to red (1.8 ns) to illustrate short (blue) and long (red) fluorescence lifetimes with the merged images and the corresponding phasor plots of the fluorescence lifetime of Fibril-Atto647 seeds at indicated timepoints after seeding. Scale bar, 10 μm. **b** Histograms of Fibril-Atto647 seeds showing the relative abundance (rel. abundance) of pixels with a certain fluorescence lifetime at each timepoint, averaged over all acquired images (n = 15 for each timepoint, mean ± SEM). Fibril-Atto647 seeds with longer fluorescence lifetimes accumulate in a time-dependent manner. **c** The mean fluorescence lifetime (τ) of Fibril-Atto647 over time. Data are shown as mean ± SEM. Statistical analysis was done using one-way ANOVA with Dunnett's multiple comparison test. **d** The ROI drawn around α-SynA53T-YFP foci (dotted white line) contains Fibril-Atto647 seed species with an intermediate lifetime (light blue) compared to the overall distribution of lifetimes (from dark blue to red). Scale bar, 10 μm.

resulting in a decrease in foci formation (Fig. 6e, f, Supplementary Fig. 7e, f). In contrast, DNAJB1 KD did not attenuate the processing of the F91 seeds and the formation of endogenous α-SynA53T-YFP foci (Fig. 6g–i, Supplementary Fig. 8e, f). Thus, F91 may not be recognized by DNAJB1 or may not be efficiently disaggregated by the HSP110/HSP70 system. Of note, DNAJA2 KD had no effect on the fluorescence lifetime and increases the seeding efficiency for both F65 and F91 (Supplementary Fig. 7g–i, 8g–i). These results suggest that the structural arrangement of α-Syn polymorphs impacts the processing ability of different protein clearance pathways.

Taken together, the broad distribution of fluorescence lifetimes appears to be attributable to the activity of multiple quality control pathways targeting aggregated proteins for degradation or disaggregation. Furthermore, the processing of α-Syn seeds by quality control pathways may produce fibrillar species with increased seeding capacity, which could accelerate the progressive spreading of pathogenic proteins.

## Discussion

The gradual accumulation and intercellular spreading of misfolded proteins is intimately linked to the development and propagation of neurodegenerative diseases. The ability of amyloidogenic proteins to form aggregates with different conformations might account for heterogeneity regarding pathology and progression[34,35]. The cellular milieu also seems to have a crucial influence[6]. Therefore, it is of great interest to investigate the role of amyloid conformation on seeded propagation in the cellular context. Here, we describe a non-invasive fluorescence-based microscopy technique that enabled us to study the dynamics and molecular events of seeded α-Syn aggregation in a biosensor cell line to unprecedented detail. α-Syn polymorphs exhibit distinct signature fingerprints in FLIM, allowing the discrimination of conformational variants. Our data further establish a mechanistic link between amyloid processing by the cellular protein quality control machinery and seeding capacity and provide a possible basis for the correlation between the age-related decline of the proteostasis network and the progression rate of various synucleinopathies.

By using sequential multi-channel FLIM, we could visualize and distinguish both the endogenous aggregation of α-Syn and the fate of the exogenous α-Syn fibrillar polymorphs during the seeding process.

While seeded aggregation did lead to a shorter fluorescence lifetime signal of the endogenous α-SynA53T-YFP reporter, the characteristic fluorescence lifetimes of the respective seeds were not replicated (Fig. 1). This was unexpected as amyloid propagation is generally thought to involve templated incorporation of monomers at the ends of the filaments, thereby preserving the conformation of the seed. Two reasons could explain why endogenous α-SynA53T-YFP showed minimal variation when inoculated with different α-Syn polymorphs in FLIM. First, α-SynA53T-YFP might not be able to adopt the exact conformation of the seed due to the specific conditions of the cellular environment, which substantially differ from the conditions the seeds

were made in regarding salt concentration, pH, etc.[1,36]. Second, the attached fluorophore could prevent α-SynA53T-YFP from adopting the exact conformation of the seed[37,38]. In both cases, the observed seeded aggregation would result primarily from secondary nucleation events rather than from a templated addition of monomers at fibril ends[39]. Thus, the conformation of fibrils formed by secondary nucleation events would be predominantly determined by solution conditions and intrinsic structural preferences of α-SynA53T-YFP rather than by the seed conformation[40].

Direct visualization of α-Syn fibrillar polymorphs labeled with Atto647 revealed that they exhibited differences in their fluorescence lifetimes. In this case, fluorophores are attached to lysine residues exposed at the surfaces of the fibrils after assembly and therefore do not affect the conformation of the distinct polymorphs. F65, for example, had the shortest initial fluorescence lifetime compared to any of the other polymorphs, while Ribbons showed relatively long fluorescence lifetimes despite their amyloid nature (Fig. 2). These differences could be caused by different positions of the attached fluorophores. We indeed recently demonstrated that the lysine residues within the distinct amyloid structures of the polymorphs are differently exposed to the solvent[27]. Fluorophores that are closer to the core of the fiber would be more quenched resulting in shorter lifetimes, while fluorophores that are closer to the N- or C-terminal end could move more freely resulting in longer lifetimes. Although this renders some polymorphs invisible in FLIM and precludes their analysis, it indirectly reflects differences in the amino acid residues that constitute the core of the fibers or the stacking of the individual monomers, and thus enables the discrimination of conformational variants. It would be interesting to compare the different α-Syn polymorphs in two-photon FLIM, which does not require labeling to examine whether this approach also allows their differentiation[41].

FLIM also allows detection of species that are formed during the seeding process. This technique revealed cellular processing of the α-Syn polymorphs over time as demonstrated by a gradual broadening of their fluorescence lifetimes, ranging from very short lifetimes, around 0.28 ns, to longer lifetimes at around 1.8 ns in the phasor plot (Fig. 4, Supplementary Fig. 4).

We show that processing of fibrillar α-Syn appears to be mediated by multiple protein clearance pathways rather than a single one (Figs. 5 and 6). The HSP110/HSP70 disaggregation system attempts to dissolve fibers thereby releasing monomeric α-Syn and generating smaller fragments[32]. A recent study showed that disaggregation of α-Syn fibrils in vitro leads to a shift of the fluorescence lifetime distribution towards longer lifetimes in FLIM[41]. Accordingly, inhibition of the disaggregation activity in our cell culture model was reflected by a decrease in fiber lifetimes in FLIM accompanied by longer fluorescence lifetimes 24 h after seeding (Fig. 5e, f). The same pattern was seen by blocking proteasomal degradation, which is consistent with observations that the proteasome can also fragment and disassemble α-Syn fibrils[42,43]. In addition, larger cytosolic α-Syn assemblies are known to be directed to the autophagy-lysosomal pathway for degradation[44], and overloading of this route could also lead to the

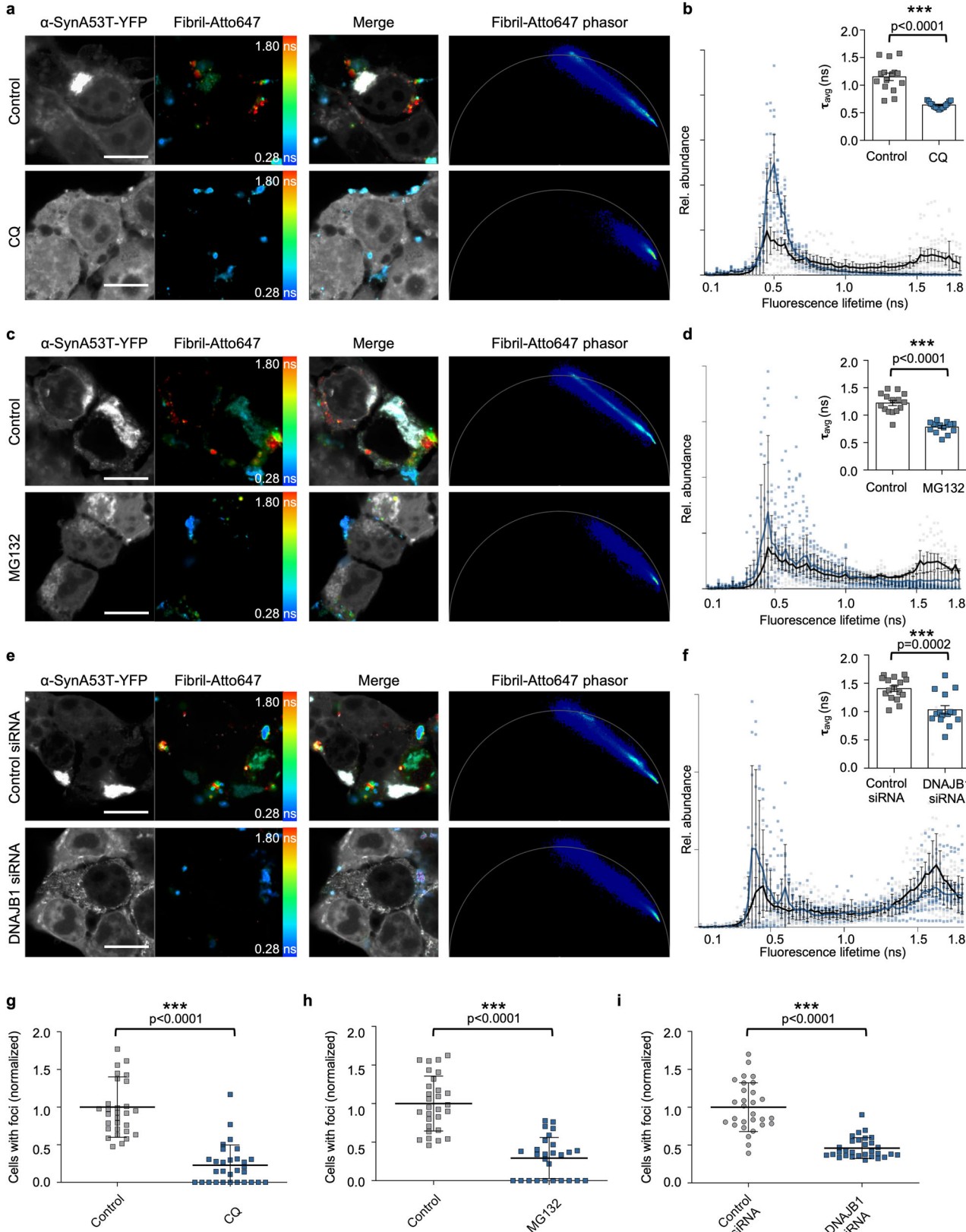

release of smaller amyloidogenic particles[45–47]. Furthermore, it is known that amyloid seeds are proteolytically processed by various cytosolic or lysosomal proteases after their uptake into the cell[48]. The cleavage of the N- and C-terminal fragments and attached fluorophores, might result in fewer intermolecular interactions of fluorophores. Whether and to what extent such proteolytic digestion contributes to the observed increase in fluorescence lifetimes, needs to be further investigated.

FLIM further revealed that structural differences in α-Syn polymorphs affected their susceptibility to different cellular aggregate clearance pathways, which could have implications in the different pathologies observed in synucleinopathies[49]. It

**Fig. 5 Protein clearance machinery augments α-Syn seeding.** Fluorescence intensity images of α-SynA53T-YFP and Fibril-Atto647 seeds pseudo colored from blue (0.28 ns) to red (1.8 ns) to illustrate short (blue) and long (red) fluorescence lifetimes with the merged images and the corresponding phasor plots of the fluorescence lifetime of Fibril-Atto647 seeds upon treatment with chloroquine (CQ) (**a**), MG132 (**c**), or DNAJB1 siRNA (**e**). Scale bar, 10 µm. **b**, **d**, **f** Histograms of Fibril-Atto647 seeds showing the relative abundance (rel. abundance) of pixels with a certain fluorescence lifetime (ranging from 0 to 1.8 ns), averaged over all acquired images ($n = 15$ for each condition, mean ± SEM), including the mean fluorescence lifetime ($\tau$) of Fibril-Atto647 upon indicated treatment. Data are shown as mean ± SEM. Statistical analysis was done using unpaired t-test. **g–i** Quantification of cells with visible foci. ($n = 30$, mean + SEM). Statistical analysis done using a Mann-Whitney test. Inhibition of the proteasome and autophagy, and silencing of the HSP110/HSP70 disaggregation machinery by DNAJB1 siRNA partially blocks the processing of the initial Fibril-Atto647 seeds, which reduces α-Syn seeding.

would be of interest to investigate seeded aggregation of α-Syn in other cell types, such as neurons or oligodendrocytes, to assess whether the processing of fibrillar α-Syn is also differentially affected by the cellular milieu[6]. Future studies using patient-derived α-synuclein conformers in more disease-relevant cell types may reveal potential disease-specific members of the proteostasis network that influence the seeded aggregation of α-Syn, which could explain the heterogeneity of synucleinopathies and pave the way toward disease-specific therapeutics[3].

However, regardless of the particular degradation machinery involved, processing of α-Syn fibers by cellular clearance pathways generally yielded species with high seeding capacity that enhanced aggregation of endogenous α-Syn (Figs. 5, 6, Supplementary Figs. 5, 7, 8). Using the phasor approach to FLIM demonstrated that the endogenous α-Syn aggregates mainly formed around α-Syn seeds with intermediate fluorescence lifetimes, not the unprocessed α-Syn seeds with the shortest fluorescence lifetimes (Fig. 4d). Hence, the observed processing seems to promote seeded aggregation.

To the best of our knowledge, we show for the first time in situ that processing by the cellular proteostasis network does generate distinct α-Syn species with high seeding capacity. Our data offer an explanation for the conundrum that is increasingly observed with protein quality control pathways, as being generally protective and mediating the degradation of toxic protein aggregates, but sometimes having the opposite effect[50,51]. Indeed, there is mounting evidence that their activity can promote amyloid propagation under certain circumstances[30,42,52].

In our model system, the clearance pathways are likely overloaded by the addition of large amounts of exogenous seeds, probably leading to slower or incomplete disaggregation and/or degradation. This seems to favor the accumulation of intermediates that exhibit high seeding propensity[47,53]. While this is an artificial system, the observation nevertheless has physiological relevance, as the capacity of the protein quality control system declines with age and to different extents in different tissues[54,55]. In particular, cellular degradation pathways such as the ubiquitin-proteasome system or the autophagy-lysosome pathway become increasingly overwhelmed during aging[56,57]. This overload and associated accumulation of intermediate, highly seeding competent amyloid species could be a reason for the progressive spreading of disease pathology in neurodegenerative diseases.

## Methods

**In vitro aggregation of recombinant α-Syn.** Human full-length wild-type human α-Syn was generated in *E. coli* BL21 DE3 CodonPlus cells (Agilent Technologies) and purified as described[58]. To generate the fibrillar polymorphs Fibrils, Ribbons, F65 and F91, full-length monomeric α-Syn (250 µM) was incubated in 20 mM Tris–HCl, pH 7.5, 150 mM KCl; in 5 mM Tris–HCl, pH 7.5; in 20 mM MES pH 6.5 and in 20 mM KPO4, 150 mM KCl, respectively, at 37 °C under continuous shaking in an Eppendorf Thermomixer set at 600 r.p.m for 7 days[1,26]. All assembly reactions were followed by withdrawing aliquots (10 µl) from the assembly reactions at different time intervals, mixing them with Thioflavin T (400 µl, 10 µM final), and recording the fluorescence increase on a Cary Eclipse Fluorescence Spectrophotometer (Varian Medical Systems Inc.) using an excitation wavelength = 440 nm, an emission wavelength = 480 nm and excitation and emission slits set at 5 and 10 nm, respectively. The resulting α-Syn fibrillar polymorphs were assessed by Transmission Electron Microscopy (TEM) after adsorption of the

fibrils onto carbon-coated 200 mesh grids and negative staining with 1% uranyl acetate using a Jeol 1400 transmission electron microscope. The fibrillar polymorphs were next labelled by addition of one molar equivalents of Atto647 NHS-ester (#AD 647-35, ATTO-Tec GmbH) fluorophore in DMSO. The mix was incubated for 1 h at room temperature. The unreacted fluorophore was removed by two centrifugations at 15,000 *g* for 10 min and resuspensions of the pellets in PBS. The fibrillar polymorphs were next fragmented to an average length of 42–52 nm by sonication for 20 min in 2 mL Eppendorf tubes in a Vial Tweeter powered by an ultrasonic processor UIS250v (250 W, 2.4 kHz; Hielscher Ultrasonic) after assembly[59]. Monomeric α-Syn (250 µM) in PBS was labeled by addition of one molar equivalent of Atto-647 NHS-ester for 2 h on ice. The unreacted dye was removed by size exclusion chromatography on NAP-5 columns. Monomeric α-Syn-Atto-647 was aliquoted (10 µl per 0.5 ml Eppendorf tube). Flash frozen in liquid nitrogen and stored at −80 °C until use.

**Cell culture.** Cells were cultured in DMEM containing high glucose, GlutaMAX Supplement, and pyruvate (Gibco), supplemented with 10% FBS (Gibco) and 1x Penicillin-Streptomycin (Gibco) at 37 °C and 5% CO2. Regular mycoplasma tests were performed (GATC Biotech). The HEK293T cell line expressing α-SynA53T-YFP was kindly provided by Marc Diamond, University of Southwestern Texas. This biosensor cell line was shown to be highly sensitive in detecting a variety of different fibrillar α-Syn species[60–62].

**Liposome-mediated seeded aggregation of cells and treatments.** Cells were seeded with different preformed α-Syn fibrils. Cells were seeded on coverslips coated with Poly-L lysine (Invitrogen) in 24-well plates in Opti-MEM Reduced Serum Medium, GlutaMAX Supplement (Gibco). 24 h later, polymorphs were combined with Opti-MEM to a concentration of 2 µM. Lipofectamine2000 (Invitrogen) was mixed with Opti-MEM (1:20 dilution) and incubated for 5 min. Lipofectamine mixture was then added 1:1 with α-Syn polymorphs and incubated for 20 min and then added to the cells to a final concentration of 100 nM. Labeled monomers of α-Syn were seeded as above with final concentration of 1 µM. For seeding longer than 12 h, the media was exchanged for complete DMEM media. The cells were fixed in 4% PFA in PBS for 10 min. After washing 3x in PBS, the cells were mounted for imaging. For fluorescence lifetime imaging on Fibrils-Atto647 with no cells, Fibrils were incubated with lipofectamine as well and then seeded and fixed in the same manner as listed above.

MG132 (ThermoScientific) was dissolved in ethanol (EtOH) and chloroquine (Sigma) dissolved in H2O and were stored at −80 °C until use. For proteasomal inhibition experiments, cells were treated with MG132 at a working concentration of 5 µM for 4 h before seeding with Fibrils for 12 h. For autophagy inhibition experiments cells were treated with chloroquine at a working concentration of 40 µM for 12 h before seeding with Fibrils for 12 h. After fiber seeding for 12 h, the cells were fixed and then imaged.

For knock down of DNAJB1 and DNAJA2 treatment, ON-Target plus SMART Pool Human DNAJB1, ON-Target plus SMART Pool Human DNAJA2 and ON-Target plus Control Pool were purchased from ThermoScientific. Due to the long experimental timeframe, cells were seeded in Opti-MEM with 10%FBS. siRNA was diluted in siRNA buffer (Final concentration 20 nM) before mixing with Dharmafect for 20 min prior to being added to cells. After 48 h knockdown, media was changed to Opti-MEM alone to seed with polymorphs for 24 h as described above.

**Confocal Imaging.** Cells were imaged with a Leica TCS SP8 STED 3X microscope (Leica Microsystem, Germany) equipped with 470–670 nm white laser with a photomultipler tube and Leica Application Suite X (LASX) software using a HCX PL APO 63x/1.40 Oil CS2 objective. All further processing of acquired images was performed with ImageJ software[63]. Manually quantification of cells and cells with fibrillar and spherical foci was done in FIJI[64] and the percentage of foci-containing cells was calculated.

**Fluorescence lifetime imaging microscopy.** For FLIM imaging, Leica SP8 FAL-CON FLIM with a time-correlated single photon counting (TCSPC) module was used (Leica microsystem, Germany). A pulsed, white light laser provided excitation at 470–670 nm and a repetition rate of 80 MHz. Images were acquired with 100X objective (HC PL APO 100x/1.40 STED White Oil) with zoom was applied up to

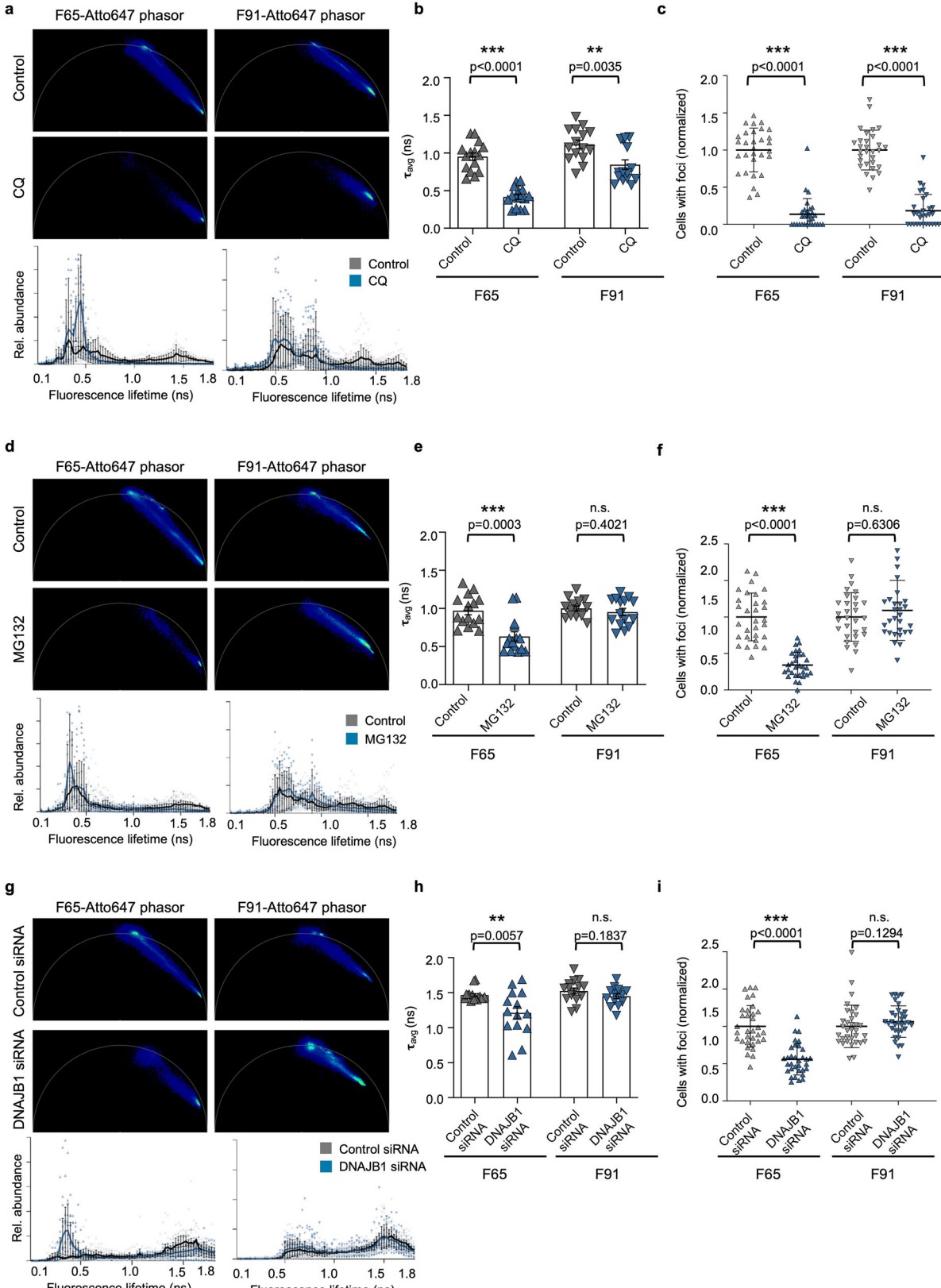

4-fold. Default FLIM settings for 514 nm and 633 nm excitation were applied for α-SynA53T-YFP and α-Syn-Atto647 polymorphs, respectively with two separate Hybrid GaAsp detectors in sequential mode (Leica microsystem, Germany). Data for FLIM were acquired until at least 1500 counts were collected in the brightest pixel of the image. Photons per laser pulse were kept under 0.5 to avoid photon pile-up. TCSPC images were analyzed using FALCON FLIM (Leica) from which lifetime histograms and phasor plots were generated. The weighted mean fluorescence lifetime was also determined using FALCON FLIM (Leica) and was fitted using bi-exponential decay.

**SDS-Page and immunoblotting**. Cells were centrifuged to pellet followed by lysis in lysis buffer (10 mM Tris, 100 mM NaCl, 0.2% Triton X-100, 10 mM EDTA) on ice for 20 min. The lysates were transferred into fresh Eppendorf tubes and

**Fig. 6 Differential processing of α-Syn polymorphs by the protein clearance pathways.** Phasor plots of the fluorescence lifetime of F65-Atto647 and F91-Atto647 seeds upon cells treatment with chloroquine (CQ) (**a**), MG132 (**d**), or DNAJB1 siRNA (**g**). Histograms of F65-Atto647 and F91-Atto647 showing the relative abundance (rel. abundance) of pixels with a certain fluorescence lifetime (ranging from 0 to 1.8 ns), averaged over all acquired images ($n = 15$ for each condition, mean ± SEM). **b, e, h** The mean fluorescence lifetime ($\tau$) of F65-Atto647 and F91-Atto647 upon indicated treatment. Data are shown as mean ± SEM. Statistical analysis was done using unpaired $t$-test (for F65 **b**, **h**, F91 **b**, **e**, **f**) or Mann-Whitney test (for F65 **e**). Quantification of cells with visible foci ($n = 30$, mean ± SEM). Statistical analysis was done using a Mann-Whitney test (for F65 **c**, F91 **c**, **f**, **i**) or an unpaired $t$-test (for F65 **f**, **i**). Inhibition of autophagy (**c**) and the proteasome (**f**), and silencing of the HSP110/HSP70 disaggregation machinery by DNAJB1 siRNA (**i**) partially blocks the processing of F65-Atto647 seeds, which reduces α-Syn seeding. In contrast, only inhibition of autophagy partially blocks the processing of F91-Atto647 seeds.

centrifuged (1000 $g$ for 1 min at 4 °C) in a tabletop centrifuge. The protein concentration was determined using protein assay dye reagent concentrate (Bio-Rad). Proteins were separated under denaturing conditions by SDS–PAGE and transferred onto a PVDF membrane (Carl Roth) by standard wet blotting protocols. Samples were probed with mouse monoclonal anti-HSP40/Hdj1 (1:1000, 2E1, Enzo) or monoclonal anti-DNAJA2 (1:1000, ab147216, Abcam) primary antibody. Anti-GAPDH antibody (1:5000, clone GAPDH-71.1, Sigma-Aldrich) was used as loading control. Alkaline phosphatase (AP)-conjugated anti-mouse IgG secondary antibodies (Vector Laboratories) were used for subsequent ECF-based detection (GE Healthcare).

**Statistics and reproducibility**. GraphPad Prism software was used to create graphs and to analyze the data. Data are presented as mean ± standard error of the mean (SEM). For each data set, $p$ values and statistical test applied are indicated in the corresponding figure legend with the following significance levels: not significant (n.s.), $p \geq 0.05$; *$p \geq 0.05$; **$p \geq 0.01$; and ***$p \geq 0.001$. All data were obtained from 3 independent biological repeats. Data sets were tested for normality using the Shapiro-Wilk test and the appropriate parametric or nonparametric test was employed. Individual results are provided in the accompanying Supplementary Data 1–13.

**Reporting summary**. Further information on research design is available in the Nature Portfolio Reporting Summary linked to this article.

## Data availability

All source data underlying the figures and graphs, as well as details in the statistical analysis, are provided in the accompanying Supplementary Data 1–13. Other relevant data and additional information are available upon request from the corresponding author.

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

## Acknowledgements

We thank Tracy Bellande for technical assistance and Holger Lorenz from the ZMBH Imaging Facility for his technical support and constructive comments on the manuscript. This work also benefited from the microscopy platform of I2BC (CNRS UMR9198, Gif-sur-Yvette, France). We are also grateful to Marc Diamond for sharing his HEK293 α-SynA53T-YFP biosensor cell line. This is an EU Joint Programme - Neurodegenerative Disease Research (JPND) project (PROTEST-70). This project is supported through the following funding organizations under the aegis of JPND - www.jpnd.eu: France, Agence National de la Recherche (ANR, ANR-17-JPCD-0005-01 to R.M.); Germany, Bundes-ministerium für Bildung und Forschung (01ED1807B to C.N-K.). Funding was also provided by the Fondation pour la Recherche Medicale (contract DEQ. 20160334896), and France Parkinson Association.

## Author contributions

Conceptualization, J.T. and C.N.-K.; Methodology, J.T., A.A., S.D.-A., R.M., and C.N.-K.; Investigation, J.T., S.D.-A.; Formal Analysis, J.T.; Resources, A.A., R.M.; Writing - Original Draft, J.T., C.N.-K.; Writing - Review and Editing, J.T., R.M., and C.N.-K.; Supervision, R.M., and C.N.-K.; Visualization, J.T.; Funding Acquisition, R.M., and C.N.-K.

## Funding

## Competing interests

The authors declare no competing interests.
