## [Peer Review File · Communications Biology]

Reviewers' comments:

Reviewer #1 (Remarks to the Author):

In this work, Tittelmeier and co-worker report an original method to follow the α -synuclein aggregation directly in cell using FLIM. The most interesting point is that the analysis of the mean fluorescent lifetime distribution can inform on the polymorphism of aggregates. The manuscript is clear and the discussion is supported by convincing results. However, several points should be clarified.

The FLIM involved the label of the α -syn, here with YFP and ATTO647. The mean fluorescent lifetime of these dyes grafted and not on the α -syn should be given as well as different stage aggregation of aggregation (with THT control) without cell.

The impact of YFP and ATTO647 affect the dynamic and the morphology of α -syn should be evaluated and discussed.

It is possible to effect a fluorescent lifetime to an aggregate morphology ?

For the phasor image correspondence colour/ value of mean fluorescence lifetime should be given.

Reviewer #2 (Remarks to the Author):

Tittelmeier et al. used phasor approach of fluorescence lifetime imaging microscopy (FLIM) to study the aggregation properties of and seeding dynamics of different α -Syn fibers in HEK 297T cell lines. Aggregation was assessed as decreased fluorescence lifetime of α -SynA53T-YFP. Seeding capacity was assessed with different α -Syn-Atto647 polymorphs. Interestingly, cellular clearance pathways yielded fibrillar species with higher fluorescence lifetime (less processed) but increased seeding capacity. Previous studies used phasor approach of FLIM to study the aggregation of proteins (e.g. <https://doi.org/10.1016/j.jcis.2020.03.107>), and FLIM has been used to study α -Syn oligomerization (<https://faseb.onlinelibrary.wiley.com/doi/10.1096/fj.05-5422com>). Thus, as a method paper, the novelty of this study is limited. However, this study adds to the proof of principle application of these methods as well as the biology by 1) distinguish the seeding and aggregation property of different α -Syn polymorphs; and 2) showing how cellular clearance pathways regulate fibrillar species as well as aggregate formation. The study is thus of interest to the field of protein aggregation and neurodegeneration, if the major concerns can be addressed.

Major concerns

1. Quantification and statistical analysis are lacking, and hence the conclusions are not well supported. Below are a few examples.

a. The authors conclude that FLIM is a powerful tool to distinguish different amyloid structures.

Without quantification and statistical analysis, it is unknown that whether the fluorescence lifetimes of different polymorph are distinguishable.

b. Figure 1c, the phasor plot of Ribbon treated α -Syn-A53T-YFP seems similar to control and different from other polymorph treated ones, inconsistent with the statement 'no robust difference in lifetimes of α -Syn-A53T-YFP upon seeding with distinct α -Syn-A53T-YFP polymorphs could be observed' (line 112-113). Please quantitate and perform statistical analysis.

c. Figure 2., quantification and statistical analysis are needed to compare the lifetime properties of α -Syn-Atto647 from all conditions. Is the lifetime distribution of Ribbon significantly different from α -Syn monomers?

d. Figure 4 and 5., please perform statistical analysis for lifetime distribution between control and treatment groups.

2. It is unclear what we learn in a cell line can generalize to neurons or disease conditions. Please include data from neurons, or at least discuss this limitation.

Specific comments:

1. The study uses a mixture of α -Syn-Atto647 fibrils and α -SynA53T-YFP in most experiments. Please discuss whether this mixture is clinically relevant, and why A53T is relevant to use here. Do wild type α -Syn-YFP display the same properties?
2. Figure 1., please include histograms of lifetime distribution.
3. Figure 2. Why would monomeric α -Syn display lifetimes between 1.5ns and 1.8ns. Should we expect it to be entirely 1.8ns?
4. Supplement Figure 1D, Ribbon fiber is not able to induce significantly more foci in cells, consistent with that the phasor plot of Ribbon treated α -Syn-A53T-YFP that looks like control in Fig. 1C. Please describe and discuss.
5. Supp. Figure 3C shows lifetime change of seeding fibers is caused purely by cellular processes without interactions with native α -Syn. This is important. Please include it in main figures.
6. The study should cite more previous papers, including but not limited to the 2 mentioned in the opening paragraph.

Reviewer #3 (Remarks to the Author):

This study describes the use of Fluorescence lifetime imaging microscopy (FLIM) combined with phasor analysis to monitor the properties of seeded α -synuclein aggregates in mammalian cells. The study shows convincingly that the phasor analysis is suited to identify differences in the *in vivo* properties of α -synuclein polymorphs, and provides evidence that amyloid seeds are remodeled *in vivo* over time. The authors attempt to include some mechanistic studies by evaluating the effects of lysosomal, proteasomal and chaperone inhibition on the properties of the α -synuclein seeds. However, the effects on the FLIM properties are generally quite subtle, and leave the reviewer wonder about what new information we learn from this analysis. Altogether, this study is very descriptive and provides little new insights into α -synuclein aggregation *in vivo*.

Major comments:

- 1) Although the authors conclude from their data that highly seeding competent degradation products are produced from some of their polymorphs, there is no direct evidence provided that there are more seeding competent species formed.
- 2) The effects of inhibiting the proteasome/DnaJ according to FLIM are quite subtle especially given that the controls themselves seem highly variable
- 3) What is the physiological consequence of cells having more or fewer foci? Does the size of the foci change?

Dear Reviewers,
Thank you for the thorough and insightful evaluation of our manuscript. We have carefully
addressed your remarks and submitted a revised manuscript for reconsideration. The changes
we made are labeled in red. Detailed responses to each of your concerns are listed under the
individual comments.

We believe that the new data strengthen our initial findings and conclusions about FLIM
being a powerful tool that can distinguish different amyloid structures and can be used to
monitor the dynamic process of amyloid remodeling by the cellular environment. We
therefore hope that you and the Editor will find our manuscript suitable for publication in
Communications Biology.

Sincerely,
Carmen Nußbaum-Krammer

Reviewers' comments:

Reviewer #1

In this work, Tittelmeier and co-worker report an original method to follow the α -synuclein
aggregation directly in cell using FLIM. The most interesting point is that the analysis of the
mean fluorescent lifetime distribution can inform on the polymorphism of aggregates. The
manuscript is clear and the discussion is supported by convincing results. However, several
points should be clarified.

We thank the reviewer for the overall positive feedback on the manuscript. We have
addressed your comments below.

1: The FLIM involved the label of the α -syn, here with YFP and ATTO647. The mean
fluorescent lifetime of these dyes grafted and not on the α -syn should be given as well as
different stage aggregation of aggregation (with THT control) without cell.

The fluorescence lifetime is an intrinsic characteristic of a fluorophore. According to the
fluorophore database FBbase, the fluorescent lifetime of enhanced YFP is 3.1 ns
(<https://www.fpbase.org/protein/eyfp/>). However, many different aspects can affect the
lifetime of fluorophores, for example attaching a protein. When YFP is fused to α -Syn and in
a cellular environment, its lifetime is around 2.9 ns (Schierle et al. 2011). The same is true for
Atto674. The lifetime of the fluorophore alone is 2.4 ns (<https://www.atto-tec.com/ATTO-647.html?language=en>), while monomeric α -Syn-Atto647 has a lifetime of 1.7 ns (Figure 2E).
Since both fluorophores do not aggregate, a reference value for the aggregated protein itself
cannot be provided.

Therefore, we think it is more meaningful to measure the lifetimes of non-aggregated α -
SynA53T-YFP and α -Syn-Atto647 to determine a reference value for the soluble proteins in
our system and then compare it to the values obtained with the aggregated proteins.

2: The impact of YFP and ATTO647 affect the dynamic and the morphology of α -syn should
be evaluated and discussed.

Attaching a fluorophore might impact the dynamic and morphology of the respective
protein. We have added the following sentences to discuss that with respect to α -SynA53T-

YFP: “While seeded aggregation did lead to a shorter fluorescence lifetime signal of the
endogenous α -SynA53T-YFP reporter, the characteristic fluorescence lifetimes of the
respective seeds were not replicated (Fig. 1). This was unexpected as amyloid propagation is
generally thought to involve templated incorporation of monomers at the ends of the filaments,
thereby preserving the conformation of the seed. Two reasons could explain why endogenous
α -SynA53T-YFP showed minimal variation when its aggregation is triggered by different α -
Syn polymorphs in FLIM. First, α -SynA53T-YFP might not be able to adopt the exact
conformation of the seed due to the specific conditions of the cellular environment, which
substantially differ from the conditions the seeds were made in regarding salt concentration,
pH, etc. (Bousset et al. 2013; Grozdanov et al. 2019). Second, the attached fluorophore could
prevent α -SynA53T-YFP from adopting the exact conformation of the seed (Afitska et al. 2017;
Caputo et al. 2020). In both cases, the observed seeded aggregation would result primarily from
secondary nucleation events rather than from a templated addition of monomers at fibrils ends
(de Oliveira and Silva 2019). Thus, the conformation of fibrils formed by secondary nucleation
events would be predominantly determined by solution conditions and intrinsic structural
preferences rather than by the seed conformation (Hadi Alijanvand, Peduzzo, and Buell 2021).”
(page 10, line 281).

We want to point out that this is less likely for the ATTO647-labelled α -Syn polymorphs,
since the fluorophore is attached to the fibers after and not during their assembly. We have
added the following sentence to indicate this: “In this case, fluorophores are attached to lysine
residues exposed at the surfaces of the fibrils after assembly and therefore do not affect the
conformation of the distinct polymorphs.” (page 11, line 297). We have also described this in
the materials and methods section.

**3: It is possible to effect a fluorescent lifetime to an aggregate morphology?**

We believe this is possible and we hope that the additional data we provided support this
claim. We show in Figure 3A that different α -Syn polymorphs display different fluorescence
lifetime signatures. To better highlight that the fluorescence lifetime of the Atto647
fluorophore is significantly affected by the aggregate morphology, we have expanded Figure
3 to show the weighted mean fluorescence lifetime with statistics (new Figure 3C).

The distinct structure of these polymorphs can be appreciated in the EM images displayed in
Supplementary Figure S1A. Moreover, they have been extensively characterized previously
(Rey et al. 2019; Shrivastava et al. 2020; Makky et al. 2016; Landureau et al. 2021). To
clarify that the polymorphs have distinct morphologies we added the following statement with
additional references (page 5, lines 113): “In addition to Fibrils, we seeded the biosensor cell
line with other structurally well-characterized fibrillar α -Syn polymorphs, F65, F91, and
Ribbons (Makky et al. 2016; Landureau et al. 2021; Rey et al. 2019; Shrivastava et al. 2020).
These polymorphs differ in the amino acids located in their core and those exposed at their
surface, resulting in a distinct fiber architecture (Supplementary Fig. 1A)(Makky et al. 2016;
Landureau et al. 2021; Rey et al. 2019; Shrivastava et al. 2020).”

**4: For the phasor image correspondence colour/ value of mean fluorescence lifetime should**
**be given.**

We thank the reviewer for this important suggestion. We have added the mean fluorescence
lifetime to all of the figures. By comparing the means we can now show that the differences
we claimed were indeed significant, which strengthens the study.

Reviewer #2 (Remarks to the Author):

Tittelmeier et al. used phasor approach of fluorescence lifetime imaging microscopy (FLIM)
to study the aggregation properties of and seeding dynamics of different α -Syn fibers in HEK
297T cell lines. Aggregation was assessed as decreased fluorescence lifetime of α -SynA53T-
YFP. Seeding capacity was assessed with different α -Syn-Atto647 polymorphs. Interestingly,
cellular clearance pathways yielded fibrillar species with higher fluorescence lifetime (less
processed) but increased seeding capacity. Previous studies used phasor approach of FLIM to
study the aggregation of proteins (e.g. <https://doi.org/10.1016/j.jcis.2020.03.107>), and FLIM
has been used to study α -Syn oligomerization
(<https://faseb.onlinelibrary.wiley.com/doi/10.1096/fj.05-5422com>). Thus, as a method paper,
the novelty of this study is limited. However, this study adds to the proof of principle
application of these methods as well as the biology by 1) distinguish the seeding and
aggregation property of different α -Syn polymorphs; and 2) showing how cellular clearance
pathways regulate fibrillar species as well as aggregate formation. The study is thus of interest
to the field of protein aggregation and neurodegeneration, if the major concerns can be
addressed.

We thank the reviewer for the comments highlighting what our study adds to the fields of
protein aggregation and neurodegeneration.

Major concerns

1. Quantification and statistical analysis are lacking, and hence the conclusions are not well
supported. Below are a few examples.

a. The authors conclude that FLIM is a powerful tool to distinguish different amyloid
structures. Without quantification and statistical analysis, it is unknown that whether the
fluorescence lifetimes of different polymorph are distinguishable.

We agree with the reviewer that quantification and statistical analysis were missing. We have
added the additional quantification of the weighted mean of the fluorescence lifetimes to
every experiment and have performed statistical analysis. We believe that this supports our
conclusions.

b. Figure 1c, the phasor plot of Ribbon treated α -Syn-A53T-YFP seems similar to control and
different from other polymorph treated ones, inconsistent with the statement 'no robust
difference in lifetimes of α -Syn-A53T-YFP upon seeding with distinct α -Syn-A53T-YFP
polymorphs could be observed' (line 112-113). Please quantitate and perform statistical
analysis.

We thank the reviewer for pointing this out. With the addition of the new analysis, we show
that Ribbons are indeed similar to the control and significantly different from the other
polymorphs. We have updated the manuscript (page 5 lines 117) to state:
“F65 and F91 polymorphs induced the formation of α -SynA53T-YFP foci (Supplementary
Fig. 1C), leading to a decreased mean fluorescence lifetime of α -SynA53T-YFP in seeded
compared to non-seeded cells similar to Fibrils (Fig. 1E-G). Ribbons had the lowest seeding
capacities of the different polymorphs in our experimental model (Supplementary Fig. 1C),
which was reflected by an unchanged fluorescence lifetime distribution and no significant
difference in the mean lifetime of α -SynA53T-YFP compared to the control (Fig. 1E-G).
Selection of short-lived α -SynA53T-YFP species on the phasor plot in Ribbon-seeded cells
revealed that they localized to foci (Fig. 1E). However, these foci also contained α -SynA53T-
YFP species with longer fluorescence lifetimes, resulting in a significantly higher mean than
the mean fluorescence lifetime of α -SynA53T-YFP species in foci seeded with Fibrils, F65 or
F91 polymorphs (Fig. 1E, Supplementary Fig. 1D). No robust difference in the mean
fluorescence lifetimes of α -SynA53T-YFP in foci seeded with Fibrils, F65 or F91 polymorphs
could be detected (Supplementary Fig. 1D). Hence, the seeded aggregation of endogenous α -
SynA53T-YFP by the addition of exogenous α -Syn polymorphs Fibrils, F65 and F91 led to
the accumulation of short-lived protein species, whereby the conformation of the added seeds
did not cause a significant difference in the respective fluorescence lifetimes.”

c. Figure 2., quantification and statistical analysis are needed to compare the lifetime
properties of α -Syn-Atto647 from all conditions. Is the lifetime distribution of Ribbon
significantly different from α -Syn monomers?

Analysis of the mean fluorescence lifetime shows there is no significant difference between
monomeric α -Syn and Ribbon polymorphs (new Figure 2F).

159 d. Figure 4 and 5., please perform statistical analysis for lifetime distribution between control
and treatment groups.

This analysis was added to the new Figure 5B, D, F and Figure 6B, E, H (previously figures 4
and 5).

2. It is unclear what we learn in a cell line can generalize to neurons or disease conditions.

Please include data from neurons, or at least discuss this limitation.

We have added the following paragraph to discuss this limitation (page 12, lines 336):

“Having limited this study to a HEK biosensor cell line, it would be of interest to investigate
seeded aggregation of α -Syn in other cell types, such as neurons or oligodendrocytes to assess
whether the processing of fibrillar α -Syn is also differentially affected by the cellular milieu
(Peng et al. 2018). Future studies using patient-derived α -Syn conformers in more disease-
relevant cell types may reveal potential disease-specific members of the proteostasis network
that influence the seeded aggregation of α -Syn, which could explain the heterogeneity of
synucleinopathies and pave the way toward disease-specific therapeutics (Hoppe, Uzunoğlu,
and Nussbaum-Krammer 2021).

However, regardless of the exact degradation machinery involved, processing of α -Syn fibers
by cellular clearance pathways generally yielded species with high seeding capacity that
enhanced aggregation of endogenous α -Syn (Fig. 5, 6, Supplementary Fig. 5, 6, 7)".
Of note, we did not investigate neuron-specific pathways or genes. Autophagy and the
ubiquitin-proteasome system are the two major proteolytic systems in all eukaryotic cells, and
the Hsp70 disaggregase is conserved in metazoan. While there might be some differences in
the composition of the proteostasis network between neuronal and non-neuronal cells, the
core machinery is highly conserved between all cell types, and therefore it is very likely that
our findings can be generalized to neurons or disease conditions.

Specific comments:

1. The study uses a mixture of α -Syn-Atto647 fibrils and α -SynA53T-YFP in most
experiments. Please discuss whether this mixture is clinically relevant, and why A53T is
relevant to use here. Do wild type α -Syn-YFP display the same properties?

We used the biosensor cell line expressing A53T mutant α -Syn because it is more sensitive to
the addition of exogenous seeds than cells expressing WT α -Syn. This is based on the
description in the original study that established this biosensor cell line, which states: "In the
HEK cells expressing α -syn140-YFP, we found 25–30% of the cells developed aggregates
upon exposure to 30 nM α -syn140*A53T fibrils, whereas over 50% of the cells expressing α -
syn140*A53T-YFP exhibited aggregates in the presence of the fibrils (Fig. 2). Based on these
findings, we chose the α -syn140*A53T-YFP cells for further study (Woerman et al. 2015)."
Moreover, this cell line was successfully used to detect fibrillar material of different α -Syn
mutants (Boyer et al. 2019; 2020; Woerman et al. 2015). We have added this information to
the materials and methods section (page 13, line 400) "This biosensor cell line was shown to
be highly sensitive in detecting a variety of different fibrillar α -Syn species". As to the
fibrillar polymorphs we used, they have been shown to trigger different pathologies in vivo
(Peelaerts et al. 2015; Rey et al. 2019) and to seed the aggregation of α -Syn to different
extents in vitro (Shrivastava et al. 2020).

2. Figure 1., please include histograms of lifetime distribution.

We have moved the histograms from Supplementary Figure 1 to the main Figure 1C and F.

3. Figure 2. Why would monomeric α -Syn display lifetimes between 1.5ns and 1.8ns. Should
we expect it to be entirely 1.8ns?

In an undisturbed environment in a well-defined buffer solution, we would expect it to be a
single value of 1.8 ns. However, in our microscopy setup, we are not able to image
monomeric α -Syn-Atto647 in a test tube. We have to transfect it into cells instead. Since the
local concentration of a monomeric protein is significantly lower than that of an aggregate, it
is much harder to detect than aggregated protein. Therefore, to detect monomeric α -Syn-
Atto647, we transfected 10x more protein (1 μ M of the monomer compared to 100 nM used
for the respective polymorphs), which might favor intermolecular interactions. Also, α -Syn is
able to form tetrameric forms and interact with membranes (Bartels, Choi, and Selkoe 2011;
Musteikytė et al. 2021). This could all affect the lifetime of the attached fluorophore.
However, this is pure speculation. After statistical analysis, we now state the mean fluorescent
lifetime of the monomer in the cellular environment (page 6 line 152).

4. Supplement Figure 1D, Ribbon fiber is not able to induce significantly more foci in cells,
consistent with that the phasor plot of Ribbon treated α -Syn-A53T-YFP that looks like control
in Fig. 1C. Please describe and discuss.

The reviewer raises an important issue that is also related to their point 1b. Ribbons show a
similar fluorescence lifetime as the control, as they are not able to induce significantly more
foci in cells compared to the control. We have updated the manuscript to state the following
(page 5, line 120) “Ribbons had the lowest seeding capacities of the different polymorphs in
our experimental model (Supplementary Fig. 1C), which was reflected by an unchanged
fluorescence lifetime distribution and no significant difference in the mean lifetime of α -
SynA53T-YFP compared to the control (Fig. 1E-G). Selection of short-lived α -SynA53T-
YFP species on the phasor plot in Ribbon-seeded cells revealed that they localized to foci
(Fig. 1E). However, these foci also contained α -SynA53T-YFP species with longer
fluorescence lifetimes, resulting in a significantly higher mean than the mean fluorescence
lifetime of α -SynA53T-YFP species in foci seeded with Fibrils, F65 or F91 polymorphs (Fig.
1E, Supplementary Fig. 1D).”

5. Supp. Figure 3C shows lifetime change of seeding fibers is caused purely by cellular
processes without interactions with native α -Syn. This is important. Please include it in main
figures.

We agree with the reviewer that this is an important control and have moved data from
Supplementary Figure 3C to the main Figure 3C.

6. The study should cite more previous papers, including but not limited to the 2 mentioned in
the opening paragraph.

We have added the following citations (bold) for the manuscript to cite more papers that
apply FLIM to investigate aggregation (page 3, line 61).

Aggregation into an amyloid fiber leads to quenching of an attached fluorophore due to
compaction and crowding, thus reducing its fluorescence lifetime (Schierle et al. 2011; Chen
et al. 2016). This can be measured using fluorescence lifetime imaging microscopy (FLIM)
(Schierle et al. 2011; Chen et al. 2016; **Gallrein et al. 2021; Hardenberg et al. 2021; De**
**Luca et al. 2020**). Therefore, FLIM is increasingly used to investigate aggregation processes
in various models (Sandhof et al. 2020; Esbjörner et al. 2014; Laine et al. 2019; **Klucken et**
**al. 2006; Pigazzini et al. 2020**).

Reviewer #3 (Remarks to the Author):

This study describes the use of Fluorescence lifetime imaging microscopy (FLIM) combined
with phasor analysis to monitor the properties of seeded α -synuclein aggregates in
mammalian cells. The study shows convincingly that the phasor analysis is suited to identify
differences in the in vivo properties of α -synuclein polymorphs, and provides evidence that
amyloid seeds are remodeled in vivo over time. The authors attempt to include some
mechanistic studies by evaluating the effects of lysosomal, proteasomal and chaperone
inhibition on the properties of the α -synuclein seeds. However, the effects on the FLIM
properties are generally quite subtle, and leave the reviewer wonder about what new

information we learn from this analysis. Altogether, this study is very descriptive and
provides little new insights into α -synuclein aggregation in vivo.

Major comments:

1) Although the authors conclude from their data that highly seeding competent degradation
products are produced from some of their polymorphs, there is no direct evidence provided
that there are more seeding competent species formed.

We have actually dedicated an entire study to the detailed characterization of the specific
products generated by the Hsp70 disaggregation machinery. We have disaggregated the Fibril
polymorph with the Hsp70 disaggregation machinery and separated the liberated products by
centrifugation and tested their individual seeding capacity in our biosensor cell line (Figure
for Reviewers). The data show that disaggregation of Fibrils generates more seeding
competent species. Smaller fragments and oligomeric species isolated from the total
disaggregation reaction have a higher seeding capacity. Monomeric species isolated from the
total disaggregation reaction were not seeding competent.

This dataset is part of a comprehensive collaborative study with the laboratory of Bernd
Bukau, which includes the use of well controlled *in vitro* experiments with purified
components. The manuscript will be published separately at a later time.

In the study described here, our scope was to characterize the seeded aggregation in the
cellular environment *in situ* by using FLIM, without additional manipulations by lysis and
centrifugation. Our data show that the occurrence of fibrillar species with intermediate
fluorescence lifetimes nicely/perfectly correlates with the seeding efficiency of the
polymorphs. Whenever we significantly interfere with this processing by pharmacological or
genetic inhibition of major degradation or disaggregation pathways, there is a significant
reduction in the seeding capacity of the respective polymorphs. FLIM is able to directly
visualize these intermediate fibrillar species and shows that they preferentially colocalize with
endogenous α -SynA53T-YFP foci (Figure 4D). Moreover, we now provide quantifications
and statistical tests that strengthen our results. Therefore, we think it is valid to conclude from
these data that the products produced from the polymorphs by cellular processing exhibit
increased seeding propensity.

2) The effects of inhibiting the proteasome/DnaJ according to FLIM are quite subtle
especially given that the controls themselves seem highly variable

This is an important point that needed clarification. We performed additional quantifications
and statistical analysis of the different treatments to show that the mean fluorescence lifetime
of the seeds is significantly shorter in treated samples in comparison to the controls. This
supports our claim that that the seeds are significantly less processed and that significantly
less species with longer lifetimes are generated, which correlates with a lower seeding
capacity. We have added these data to Figure 5D, F and 6E, H and Supplementary Figure 5H,
6H, and 7H (previously Figure 4 and 5).

3) What is the physiological consequence of cells having more or fewer foci? Does the size of
the foci change?

The HEK biosensor cells used in this study are widely used to detect and quantify the
presence of seeding-competent protein species (Boyer et al. 2019; 2020; Woerman et al.
2015). These seeding-competent protein species are believed to drive disease progression
because they are able to self-propagate and correlate with cytotoxicity in patients and animal
models (Boyer et al. 2019; 2020; Woerman et al. 2015). However, HEK cells are quite robust
and there is only little physiological consequence for these cells having foci. This is likely due
to the fact that this cell line is rapidly dividing, unlike neurons in the human brain or other cell
types that are terminally differentiated. Therefore, toxic protein species do not accumulate but
are diluted during continuous cell divisions. The question regarding physiological
consequences needs to be addressed in more sophisticated model systems, such as animal
models. However, we believe that this is beyond the scope of our study, which was to
establish the use of FLIM to monitor α -Syn seeding and aggregation dynamics in the cellular
environment and to show that FLIM allows the detection of different fibrillar species in cells
*in situ*.

Regarding the size of the foci, we did observe an intriguing change in the size and shape of
the foci after knockdown of DNAJB1. We have described this observation on page 9, line
229: “Of note, while DNAJB1 KD reduced foci formation in general, we noticed an increase
in elongated foci as opposed to the typical spherical foci (Supplementary Fig. 5D zoom, 5E),
suggesting that DNAJB1 may affect not only exogenously added seeds but also endogenous
α -SynA53T-YFP aggregates.”

**References:**

- Afitska, Kseniia, Anna Fucikova, Volodymyr V. Shvadchak, and Dmytro A. Yushchenko.
2017. “Modification of C Terminus Provides New Insights into the Mechanism of α -
Synuclein Aggregation.” *Biophysical Journal* 113 (10): 2182–91.
<https://doi.org/10.1016/j.bpj.2017.08.027>.
- Bartels, Tim, Joanna G. Choi, and Dennis J. Selkoe. 2011. “ α -Synuclein Occurs
Physiologically as a Helically Folded Tetramer That Resists Aggregation.” *Nature* 477
(7362): 107. <https://doi.org/10.1038/NATURE10324>.
- Bousset, Luc, Laura Pieri, Gemma Ruiz-Arlandis, Julia Gath, Poul Henning Jensen, Birgit
Habenstein, Karine Madiona, et al. 2013. “Structural and Functional Characterization of
Two Alpha-Synuclein Strains.” *Nature Communications* 4.
<https://doi.org/10.1038/ncomms3575>.
- Boyer, David R., Binsen Li, Chuanqi Sun, Weijia Fan, Michael R. Sawaya, Lin Jiang, and
David S. Eisenberg. 2019. “Structures of Fibrils Formed by α -Synuclein Hereditary
Disease Mutant H50Q Reveal New Polymorphs.” *Nature Structural & Molecular
Biology* 26 (11): 1044. <https://doi.org/10.1038/S41594-019-0322-Y>.
- Boyer, David R., Binsen Li, Chuanqi Sun, Weijia Fan, Kang Zhou, Michael P. Hughes,
Michael R. Sawaya, Lin Jiang, and David S. Eisenberg. 2020. “The α -Synuclein
Hereditary Mutation E46K Unlocks a More Stable, Pathogenic Fibril Structure.”
*Proceedings of the National Academy of Sciences of the United States of America* 117
(7): 3592–3602. <https://doi.org/10.1073/PNAS.1917914117/-/DCSUPPLEMENTAL>.
- Caputo, Anna, Yuling Liang, Tobias D. Raabe, Angela Lo, Mian Horvath, Bin Zhang,
Hannah J. Brown, Anna Stieber, and Kelvin C. Luk. 2020. “Snca-GFP Knock-In Mice
Reflect Patterns of Endogenous Expression and Pathological Seeding.” *ENeuro* 7 (4): 1–
18. <https://doi.org/10.1523/ENEURO.0007-20.2020>.

Chen, WeiYue, Laurence J. Young, Meng Lu, Alessio Zaccane, Florian Ströhl, Na Yu,
Gabriele S. Kaminski Schierle, and Clemens F. Kaminski. 2016. "Fluorescence Self-
Quenching from Reporter Dyes Informs on the Structural Properties of Amyloid Clusters
Formed in Vitro and in Cells." *Nano Letters* 17 (1): 143–49.
<https://doi.org/10.1021/ACS.NANOLETT.6B03686>.

Esbjörner, Elin K., Fiona Chan, Eric Rees, Miklos Erdelyi, Leila M. Luheshi, Carlos W.
Bertoncini, Clemens F. Kaminski, Christopher M. Dobson, and Gabriele S. Kaminski
Schierle. 2014. "Direct Observations of Amyloid β Self-Assembly in Live Cells Provide
Insights into Differences in the Kinetics of A β (1–40) and A β (1–42) Aggregation."
*Chemistry & Biology* 21 (6): 732. <https://doi.org/10.1016/J.CHEMBIOL.2014.03.014>.

Gallrein, Christian, Manuel Iburg, Tim Michelberger, Alen Koçak, Dmytro Puchkov, Fan Liu,
Sara Maria Ayala Mariscal, Tanmoyita Nayak, Gabriele S. Kaminski Schierle, and
Janine Kirstein. 2021. "Novel Amyloid-Beta Pathology C. Elegans Model Reveals
Distinct Neurons as Seeds of Pathogenicity." *Progress in Neurobiology* 198 (March).
<https://doi.org/10.1016/J.PNEUROBIO.2020.101907>.

Grozdanov, Veselin, Luc Bousset, Meike Hoffmeister, Corinna Bliederhaeuser, Christoph
Meier, Karine Madiona, Laura Pieri, et al. 2019. "Increased Immune Activation by
Pathologic A-Synuclein in Parkinson's Disease." *Annals of Neurology* 86 (4): 593–606.
<https://doi.org/10.1002/ana.25557>.

Hadi Alijanvand, Saeid, Alessia Peduzzo, and Alexander K. Buell. 2021. "Secondary
Nucleation and the Conservation of Structural Characteristics of Amyloid Fibril Strains."
*Frontiers in Molecular Biosciences* 8 (April): 268.
<https://doi.org/10.3389/fmolb.2021.669994>.

Hardenberg, Maarten C., Tessa Sinnige, Sam Casford, Samuel T. Dada, Chetan Poudel,
Elizabeth A. Robinson, Monika Fuxreiter, et al. 2021. "Observation of an α -Synuclein
Liquid Droplet State and Its Maturation into Lewy Body-like Assemblies." *Journal of*
*Molecular Cell Biology* 13 (4): 282–94. <https://doi.org/10.1093/JMCB/MJAA075>.

Hoppe, Simon Oliver, Gamze Uzunoğlu, and Carmen Nussbaum-Krammer. 2021. " α -
Synuclein Strains: Does Amyloid Conformation Explain the Heterogeneity of
Synucleinopathies?" *Biomolecules* 2021, Vol. 11, Page 931 11 (7): 931.
<https://doi.org/10.3390/BIOM11070931>.

Klucken, Jochen, Tiago F. Outeiro, Paul Nguyen, Pamela J. McLean, and Bradley T. Hyman.
2006. "Detection of Novel Intracellular Alpha-synuclein Oligomeric Species by
Fluorescence Lifetime Imaging." *The FASEB Journal* 20 (12): 2050–57.
<https://doi.org/10.1096/FJ.05-5422COM>.

Laine, Romain F., Tessa Sinnige, Kai Yu Ma, Amanda J. Haack, Chetan Poudel, Peter Gaida,
Nathan Curry, et al. 2019. "Fast Fluorescence Lifetime Imaging Reveals the Aggregation
Processes of α -Synuclein and Polyglutamine in Aging Caenorhabditis Elegans." *ACS*
*Chemical Biology* 14 (7): 1628–36. <https://doi.org/10.1021/ACSCHEMBIO.9B00354>.

Landureau, Maud, Virginie Redeker, Tracy Bellande, Stéphanie Eyquem, and Ronald Melki.
2021. "The Differential Solvent Exposure of N-Terminal Residues Provides
'Fingerprints' of Alpha-Synuclein Fibrillar Polymorphs." *Journal of Biological*
*Chemistry* 296 (January): 100737. <https://doi.org/10.1016/J.JBC.2021.100737>.

Luca, Giuseppe De, Dirk Fennema Galparsoro, Giuseppe Sancataldo, Maurizio Leone, Vito
Foderà, and Valeria Vetri. 2020. "Probing Ensemble Polymorphism and Single
Aggregate Structural Heterogeneity in Insulin Amyloid Self-Assembly." *Journal of*
*Colloid and Interface Science* 574 (August): 229–40.
<https://doi.org/10.1016/j.jcis.2020.03.107>.

Makky, Ali, Luc Bousset, Jérôme Polesel-Maris, and Ronald Melki. 2016. "Nanomechanical
Properties of Distinct Fibrillar Polymorphs of the Protein α -Synuclein." *Scientific*
*Reports* 6 (1): 1–10. <https://doi.org/10.1038/srep37970>.

Musteikytė, Greta, Akhila K. Jayaram, Catherine K. Xu, Michele Vendruscolo, Georg
Krainer, and Tuomas P.J. Knowles. 2021. “Interactions of α -Synuclein Oligomers with
Lipid Membranes.” *Biochimica et Biophysica Acta (BBA) - Biomembranes* 1863 (4):
183536. <https://doi.org/10.1016/J.BBAMEM.2020.183536>.

Oliveira, Guilherme A.P. de, and Jerson L. Silva. 2019. “Alpha-Synuclein Stepwise
Aggregation Reveals Features of an Early Onset Mutation in Parkinson’s Disease.”
*Communications Biology* 2:1 2 (1): 1–13. [https://doi.org/10.1038/s42003-019-](https://doi.org/10.1038/s42003-019-0598-9)
[0598-9](https://doi.org/10.1038/s42003-019-0598-9).

Peelaerts, W., L. Bousset, A. Van Der Perren, A. Moskalyuk, R. Pulizzi, M. Giugliano, C.
Van Den Haute, R. Melki, and V. Baekelandt. 2015. “ α -Synuclein Strains Cause Distinct
Synucleinopathies after Local and Systemic Administration.” *Nature* 522 (7556): 340–
44. <https://doi.org/10.1038/nature14547>.

Peng, Chao, Ronald J Gathagan, Dustin J Covell, Coraima Medellin, Anna Stieber, John L
Robinson, Bin Zhang, et al. 2018. “Cellular Milieu Imparts Distinct Pathological α -
Synuclein Strains in α -Synucleinopathies.” *Nature* 557 (7706): 558–63.
<https://doi.org/10.1038/s41586-018-0104-4>.

Pigazzini, Maria Lucia, Christian Gallrein, Manuel Iburg, Gabriele Kaminski Schierle, and
Janine Kirstein. 2020. “Characterization of Amyloid Structures in Aging C. Elegans
Using Fluorescence Lifetime Imaging.” *JoVE (Journal of Visualized Experiments)* 2020
(157): e61004. <https://doi.org/10.3791/61004>.

Rey, Nolwen L., Luc Bousset, Sonia George, Zachary Madaj, Lindsay Meyerdirk, Emily
Schulz, Jennifer A. Steiner, Ronald Melki, and Patrik Brundin. 2019. “ α -Synuclein
Conformational Strains Spread, Seed and Target Neuronal Cells Differentially after
Injection into the Olfactory Bulb.” *Acta Neuropathologica Communications* 2019 7:1 7
(1): 1–18. <https://doi.org/10.1186/S40478-019-0859-3>.

Sandhof, Carl Alexander, Simon Oliver Hoppe, Silke Druffel-Augustin, Christian Gallrein,
Janine Kirstein, Cindy Voisine, and Carmen Nussbaum-Krammer. 2020. “Reducing
INS-IGF1 Signaling Protects against Non-Cell Autonomous Vesicle Rupture Caused by
SNCA Spreading.” *Autophagy* 16 (5): 878–99.
<https://doi.org/10.1080/15548627.2019.1643657>.

Schierle, Dr Gabriele S. Kaminski, Dr Carlos W. Bertoncini, Fiona T. S. Chan, Annemieke T.
van der Goot, Dr Stefanie Schwedler, Dr Jeremy Skepper, Dr Simon Schlachter, et al.
2011. “A FRET Sensor for Non-Invasive Imaging of Amyloid Formation in Vivo.”
*Chemphyschem : A European Journal of Chemical Physics and Physical Chemistry* 12
(3): 673. <https://doi.org/10.1002/CPHC.201000996>.

Shrivastava, Amulya Nidhi, Luc Bousset, Marianne Renner, Virginie Redeker, Jimmy
Savistchenko, Antoine Triller, and Ronald Melki. 2020. “Differential Membrane Binding
and Seeding of Distinct α -Synuclein Fibrillar Polymorphs.” *Biophysical Journal* 118 (6):
1301. <https://doi.org/10.1016/J.BPJ.2020.01.022>.

Woerman, Amanda L., Jan Stöhr, Atsushi Aoyagi, Ryan Rampersaud, Zuzana Krejciova, Joel
C. Watts, Takao Ohyama, et al. 2015. “Propagation of Prions Causing Synucleinopathies
in Cultured Cells.” *Proceedings of the National Academy of Sciences of the United States*
*of America* 112 (35): E4949–58. <https://doi.org/10.1073/pnas.1513426112>.

REVIEWERS' COMMENTS:

Reviewer #1 (Remarks to the Author):

The authors have made an effort to take into account the referees comments correct the manuscript. I feel that the manuscript could be accepted in present form.

Reviewer #2 (Remarks to the Author):

Tittelmeier et al. significantly improved their manuscript by adding statistical analyses of their data, including measured interpretation that match their results, and discussing the limitation of the study. I have three minor comments.

1. Most statistical analyses used unpaired T-tests. Did the authors test for normality of the data? If not, or if the data do not follow normal distribution, please use non-parametric statistical tests.
2. It is still unclear to me whether α -SynA53T-YFP is clinically relevant? Does the mutation happen in human patients? If so, please discuss and cite relevant papers
3. The writing is often unclear, contains many grammar mistakes and typo, and does not do justice to the study. The manuscript can benefit from a good writing or editing service. Here are only a few examples.

a. page 12, lines 336): Incorrect grammar

"Having limited this study to a HEK biosensor cell line, it would be of interest to investigate seeded aggregation of α -Syn in other cell types, such as neurons or oligodendrocytes to assess whether the processing of fibrillar α -Syn is also differentially affected by the cellular milieu (Peng et al. 2018).

b. typo: "pave to way forward"

"Future studies using patient-derived α -Syn conformers in more disease relevant cell types may reveal potential disease-specific members of the proteostasis network that influence the seeded aggregation of α -Syn, which could explain the heterogeneity of synucleinopathies and pave to way toward disease-specific therapeutics (Hoppe, Uzunoğlu, and Nussbaum-Krammer 2021).

c. page 5 lines 117: Unclear and long sentence, too many comparisons in a single sentence.

"F65 and F91 polymorphs induced the formation of α -SynA53T-YFP foci (Supplementary Fig. 1C), leading to a decreased mean fluorescence lifetime of α -SynA53T-YFP in seeded compared to non-seeded cells similar to Fibrils (Fig. 1E-G).

d. Unclear and long sentence.

However, these foci also contained α -SynA53T YFP species with longer fluorescence lifetimes, resulting in a significantly higher mean than the mean fluorescence lifetime of α -SynA53T-YFP species in foci seeded with Fibrils, F65 or F91 polymorphs (Fig. 1E, Supplementary Fig. 1D).

Reviewer #3 (Remarks to the Author):

The authors have addressed my concerns to a reasonable extent.

Dear Reviewer #2,

we have addressed your additional remarks and submitted a revised manuscript for reconsideration. Detailed responses to each of your concerns are listed under the individual comments.

Sincerely,
Carmen Nußbaum-Krammer

Reviewers' comments:

Reviewer #2

Tittelmeier et al. significantly improved their manuscript by adding statistical analyses of their data, including measured interpretation that match their results, and discussing the limitation of the study. I have three minor comments.

1. Most statistical analyses used unpaired T-tests. Did the authors test for normality of the data? If not, or if the data do not follow normal distribution, please use non-parametric statistical tests.

Data sets were tested for normality and results are provided in the supplementary data files. We have also added a statement to the methods to highlight the employed normality testing.

2. It is still unclear to me whether α -SynA53T-YFP is clinically relevant? Does the mutation happen in human patients? If so, please discuss and cite relevant papers

The A53T mutation is a clinically relevant missense point mutation that has been associated with autosomal dominant, early-onset PD. We have now highlight that A53T is a disease-related mutation in the description of the biosensor cell line (Line 93).

3. The writing is often unclear, contains many grammar mistakes and typo, and does not do justice to the study. The manuscript can benefit from a good writing or editing service. Here are only a few examples.

a. page 12, lines 336): Incorrect grammar

“Having limited this study to a HEK biosensor cell line, it would be of interest to investigate seeded aggregation of α -Syn in other cell types, such as neurons or oligodendrocytes to assess whether the processing of fibrillar α -Syn is also differentially affected by the cellular milieu (Peng et al. 2018).

The sentence was changed to: “It would be of interest to investigate seeded aggregation of α -Syn in other cell types, such as neurons or oligodendrocytes, to assess whether the processing of fibrillar α -Syn is also differentially affected by the cellular milieu⁶.”

b. typo: “pave to way forward”

“Future studies using patient-derived α -Syn conformers in more disease relevant cell types may reveal potential disease-specific members of the proteostasis network that influence the seeded aggregation of α -Syn, which could explain the heterogeneity of synucleinopathies and pave to way toward disease-specific therapeutics (Hoppe, Uzunoğlu, and Nussbaum-Krammer 2021).

Typo was changed to “pave the way forward”.

c. page 5 lines 117: Unclear and long sentence, too many comparisons in a single sentence. "F65 and F91 polymorphs induced the formation of a-SynA53T-YFP foci (Supplementary Fig. 1C), leading to a decreased mean fluorescence lifetime of a-SynA53T-YFP in seeded compared to non-seeded cells similar to Fibrils (Fig. 1E-G).

The sentence was changed to "Similar to Fibrils, F65 and F91 polymorphs induced the formation of α -SynA53T-YFP foci. This led to a decreased mean fluorescence lifetime of α -SynA53T-YFP in seeded compared to non-seeded cells (Fig. 1E-G, Supplementary Fig. 1C)."

d. Unclear and long sentence.

However, these foci also contained a-SynA53T YFP species with longer fluorescence lifetimes, resulting in a significantly higher mean than the mean fluorescence lifetime of a-SynA53T-YFP species in foci seeded with Fibrils, F65 or F91 polymorphs (Fig. 1E, Supplementary Fig. 1D).

The sentence was changed to: "However, these foci also contained α -SynA53T-YFP species with longer fluorescence lifetimes. Therefore, α -SynA53T-YFP foci seeded by Ribbons have a significantly higher mean fluorescence lifetime than α -SynA53T-YFP foci seeded by Fibrils, F65, or F91 polymorphs (Fig. 1E, Supplementary Fig. 1D)."